# ParoQuant: Pairwise Rotation Quantization for Efficient Reasoning LLM Inference

**Yesheng Liang**[3,†]    **Haisheng Chen**[3,‡]    **Song Han**[1,2]    **Zhijian Liu**[3]
[1]NVIDIA    [2]MIT    [3]UC San Diego
[†]Algorithm lead    [‡]System lead

https://paroquant.z-lab.ai

## Abstract

Post-training quantization (PTQ) compresses the weights and activations of large language models (LLMs) into low-precision representations to reduce memory footprint and accelerate inference. However, the presence of outliers in weights and activations often leads to large quantization errors and severe accuracy degradation, especially in recent reasoning LLMs where errors accumulate across long chains of thought. Existing PTQ methods either fail to sufficiently suppress outliers or introduce significant overhead during inference. In this paper, we propose **Pairwise Rotation Quantization** (ParoQuant), a PTQ method that combines hardware-efficient and optimizable *independent Givens rotations* with channel-wise scaling to even out the magnitudes across channels and narrow the dynamic range within each quantization group, effectively addressing the outlier issue. We further co-design the inference kernel to fully exploit GPU parallelism and keep the rotations and scaling lightweight at runtime. Under weight-only quantization, ParoQuant achieves an average **2.4%** accuracy improvement over AWQ on reasoning tasks, with less than 10% overhead. ParoQuant also matches the accuracy of state-of-the-art weight-activation quantization methods. This paves the way for more efficient and accurate deployment of reasoning LLMs.

## 1 Introduction

Large language models (LLMs) have demonstrated remarkable capabilities across a wide range of tasks. However, their massive size and large memory footprint not only incur substantial inference costs but also hinder on-device deployment. To accelerate edge deployments, weight-only post-training quantization (PTQ) converts model weights to lower-bit-width representations (*e.g.*, INT4), reducing the memory footprint during inference and thus increasing throughput in memory-bound autoregressive decoding. To accelerate inference in large-batch LLM serving, activations can also be quantized to enable matrix multiplications in lower bitwidths, further reducing computational costs.

Nevertheless, both activations and weights in LLMs possess many outliers (Dettmers et al., 2022; Xiao et al., 2023; Lin et al., 2024b), making it challenging to preserve the original precision under low-bit quantization. Most existing PTQ methods (Frantar et al., 2023; Lin et al., 2024b; Wei et al., 2023; Shao et al., 2024; Lee et al., 2024; Ashkboos et al., 2024; Chen et al., 2025; Liu et al., 2025b; Tseng et al., 2024b; Sun et al., 2025) try to mitigate the impact of outliers, yet they either incur large quantization errors due to suboptimal outlier elimination or introduce significant overhead from arithmetic-intensive computation. For example, in weight-only quantization, AWQ (Lin et al., 2024b), a widely adopted and fast quantization method, causes a **2.8% accuracy drop** of 4-bit quantized Qwen3-4B (Yang et al., 2025) on MMLU-Pro (Wang et al., 2024). In contrast, QTIP (Tseng et al., 2024b), which achieves state-of-the-art quantization accuracy, is about **30% slower** than AWQ because of the extra overhead introduced to mitigate outliers.

With the advent of reasoning LLMs (Jaech et al., 2024; Guo et al., 2025; Yang et al., 2025), we argue that *both* accuracy and efficiency are critical for practical quantization methods. Reasoning models achieve superior performance by generating a large number of chain-of-thought tokens, presenting unique challenges for quantization. On the one hand, quantization error *accumulates* at each decoding

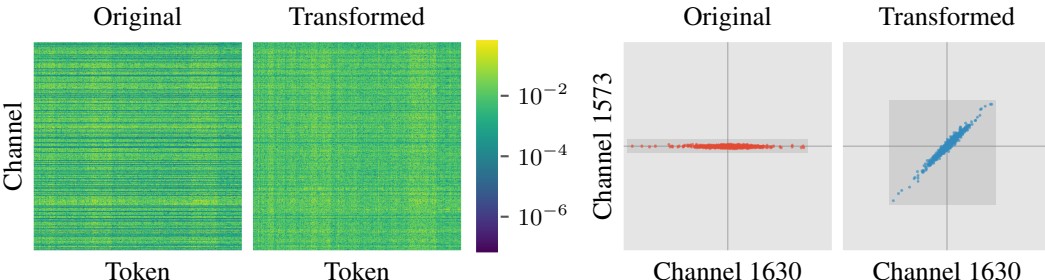

Figure 1: Effect of optimized channel-wise scaling and rotations. **Left**: Magnitude of the k_proj weight in the first layer of LLaMA-3-8B (Grattafiori et al., 2024) before and after the transform. The outlier channels have been eliminated effectively. **Right**: Scatter of two channels of the weight matrix before and after the transform. In addition to scaling, which concentrates values of the entire channel, rotations draw values from different channels closer at *each token* (clustering around the $x = y$ line).

step, which becomes particularly pronounced in long generation (Zhao et al., 2025; Liu et al., 2025a). On the other hand, the substantial computational cost of generating long sequences requires that the quantization process itself introduce negligible overhead. Thus, there is a critical need for a quantization method that achieves high fidelity with minimal extra overhead.

In this paper, we propose **Pa**irwise **Ro**tation **Quant**ization (ParoQuant), a PTQ method that combines high accuracy with minimal computational overhead, making it well-suited to reasoning LLMs. Our design rests on two key observations: (1) rotations effectively suppress outliers, and (2) a sparsely parameterized rotation can be as effective as a full rotation. Building on these insights, we introduce **scaled pairwise rotation**, a hardware-efficient and optimizable transform composed of *independent Givens rotations* and *channel-wise scaling*. Channel-wise scaling evens out the average magnitude across channels, while the pairwise (*i.e.*, Givens) rotations align the values within each channel pair at every token position, narrowing the dynamic range of each quantization group. As illustrated in Figure 1, our proposed transform makes the weights more quantization-friendly. To fully exploit the massive parallelism of modern GPUs, we further constrain the rotations to be mutually independent, a system-level design choice that keeps the impact on decoding latency minimal. Thanks to our algorithm-system co-design, under weight-only quantization, ParoQuant achieves an average **2.4%** improvement over AWQ on reasoning tasks with less than 10% extra overhead, and matches the accuracy of QTIP while being about **25% faster**. ParoQuant is also applicable for weight-activation quantization, and it matches the best existing quantization methods.

## 2 BACKGROUND AND RELATED WORK

### 2.1 LLM QUANTIZATION

Quantization is the process of converting values from high-precision to low-precision counterparts. The simple Round-to-Nearest (RTN) linear quantization with bit width $b$ can be formulated as:

$$Q(\mathbf{X}) = \text{clamp}\left(\left\lfloor \frac{\mathbf{X}}{s} \right\rceil + z, 0, 2^b - 1\right), \text{where } s = \frac{\max(\mathbf{X}) - \min(\mathbf{X})}{2^b - 1}, z = -\left\lfloor \frac{\min(\mathbf{X})}{s} \right\rceil. \quad (1)$$

In this work, we focus on *weight-only* PTQ, *i.e.*, quantizing weights of pre-trained models while keeping the activations in FP16, although our method can be extended to weight-activation quantization (Section A.3). We follow the best practices proposed by Dettmers & Zettlemoyer (2023) and adopt block-wise quantization with a given group size $g$, *i.e.*, calculating a separate $s$ and $z$ in Equation (1) for every $g$ consecutive elements along the channel dimension (*i.e.*, input dimension), instead of the whole matrix. Blocking helps to confine outliers within each group and increases overall quantization accuracy, particularly in linear quantization where quantization error is relatively large.

One major challenge of quantizing pre-trained LLMs to low bits is the presence of *outlier channels* across layers (Dettmers et al., 2022; Xiao et al., 2023; Lin et al., 2024b). They occupy the limited dynamic range of low-bit representations and cause precision loss of non-outlier elements, presenting a major challenge to PTQ. Past works have extensively studied the approaches to address the outlier issue, and the solutions can be broadly grouped into three categories: storing the outliers separately

in higher precision (Dettmers et al., 2022; Kim et al., 2024; Lee et al., 2024; Zhao et al., 2024), designing quantization algorithms suitable for non-uniform distributions (Frantar et al., 2023; Chee et al., 2023; Tseng et al., 2024a;b), and transforming weights into quantization-friendly counterparts before quantization (Lin et al., 2024b; Wei et al., 2023; Shao et al., 2024; Ashkboos et al., 2024; Lin et al., 2024a; Chee et al., 2023; Liu et al., 2025b; Tseng et al., 2024a;b; Sun et al., 2025; Malinovskii et al., 2025). Yet, it remains a key challenge to balance quantization accuracy and inference speed, as effective outlier elimination often comes at the cost of significant overhead.

## 2.2 EQUIVALENT WEIGHT TRANSFORM

Among the three outlier handling techniques discussed earlier, *transforming weights before quantization* has been widely adopted by most recent PTQ methods and has proven to be very effective. For a linear layer $\mathbf{Y} = \mathbf{XW} + \mathbf{b}$, where input $\mathbf{X} \in \mathbb{R}^{T \times C_{\text{in}}}$, weight $\mathbf{W} \in \mathbb{R}^{C_{\text{in}} \times C_{\text{out}}}$, and bias $\mathbf{b} \in \mathbb{R}^{1 \times C_{\text{out}}}$, we can apply an invertible transform $\mathbf{T}$ to the weight $\mathbf{W}$ without affecting the output:

$$\mathbf{Y} = \mathbf{XW} + \mathbf{b} = (\mathbf{XT}^{-1})(\mathbf{TW}) + \mathbf{b}. \tag{2}$$

We then quantize $\mathbf{TW}$ instead of $\mathbf{W}$. An appropriate $\mathbf{T}$ can reduce the outliers in $\mathbf{W}$ and lead to higher quantization accuracy. The inverse transform $\mathbf{T}^{-1}$ can either be applied on the fly during inference or be merged into other operators, depending on the characteristics of the transform.

Two main types of transform have been proposed in previous work: *channel-wise scaling*, where $\mathbf{T}$ is a diagonal matrix (Lin et al., 2024b; Shao et al., 2024; Wei et al., 2023), and *rotation*, where $\mathbf{T}$ is an orthogonal matrix (Chee et al., 2023; Ashkboos et al., 2024; Liu et al., 2025b; Lin et al., 2024a; Tseng et al., 2024a;b; Sun et al., 2025; Malinovskii et al., 2025). Channel-wise scaling scales each channel separately to even out the magnitude across channels and can usually be merged into preceding operators without incurring extra overhead (Lin et al., 2024b). Rotation enables cross-channel interactions that can concentrate values more effectively than channel-wise scaling (Chee et al., 2023; Liu et al., 2025b). However, rotations cannot be merged into element-wise operators like channel-wise scaling does, so they usually require online computation. This limits the application of rotations in efficient quantization algorithms, as common orthogonal transforms are computationally expensive, and it motivates the design of more efficient yet equally effective alternatives.

## 3 MOTIVATION

**Quantization Error Accumulates in Long Generation.** AWQ (Lin et al., 2024b) is a widely used weight-only quantization method and has become the *de facto* approach for INT4 quantization. It employs channel-wise scaling to minimize quantization error and causes only slight performance degradation on most tasks with limited generated tokens, without introducing any extra overhead from the transform. However, we observe that the degradation becomes more severe as the generation length grows, especially on reasoning tasks with reasoning models, where the generation length often exceeds tens of thousands of tokens. For example, the accuracy of Qwen3-4B (Yang et al., 2025) on MMLU-Pro (Wang et al., 2024) drops sharply **from 71.0 to 68.2** after being quantized to 4 bits with AWQ. This degradation occurs because quantization errors accumulate at each decoding step.

**Rotations Are Expressive but Expensive.** Rotations outperform channel-wise scaling in eliminating outliers and generally lead to lower quantization error when many outliers are present (Figure 2). However, applying arbitrary rotations requires performing matrix multiplications in FP16, which negates the efficiency gains of quantization. There are two main approaches to address this issue. SpinQuant (Liu et al., 2025b) proposes to merge the rotation matrix into the weight of the preceding linear layer so that no extra computation is needed during inference. However, in a typical decoder block, the output projection is the only linear layer that can be transformed by such mergeable rotations; other linears are preceded by element-wise operators or residual connections that cannot absorb matrix multiplications. The second approach is to restrict the orthogonal transform to a subset that can be computed efficiently on the fly. Several works adopt the Hadamard transform, a special orthogonal transform that can be computed in $\mathcal{O}(n \log n)$ time for dimension $n$ (Chee et al., 2023; Ashkboos et al., 2024; Liu et al., 2025b; Tseng et al., 2024a;b). Yet the Hadamard transform is fixed or is generated by random vectors, disregarding the unique weight distribution of each linear layer and introducing large variance (Liu et al., 2025b). Moreover, it still adds considerable overhead, making Hadamard-based quantization significantly slower ($\approx$30%) than AWQ during inference.

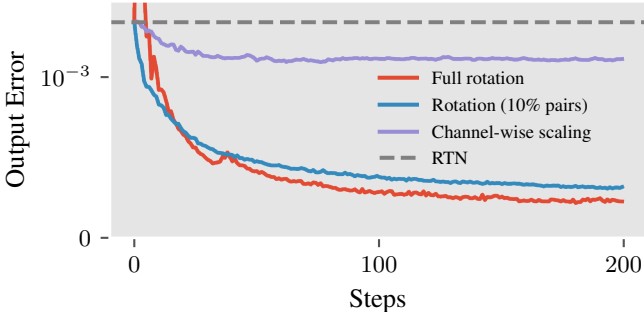

Figure 2: Loss curves from optimizing transforms to minimize quantization-induced output error ($\|\mathbf{X}Q(\mathbf{W}) - \mathbf{X}\mathbf{W}\|$) for the k_proj weight matrix in the first layer of LLaMA-3-8B. Rotations can minimize quantization error better than channel-wise scaling, and keeping the 10% most significant pairs is equally expressive as a full rotation. See Section A.2 for more details.

**Rotations Have Many Redundant Parameters.** An $n \times n$ orthogonal matrix can be decomposed into the product of at most $\frac{1}{2}n(n-1)$ Givens rotations (*i.e.*, rotations in the plane spanned by two axes), which translates to rotating all possible pairs of channels sequentially. Intuitively, rotations between an outlier channel and a normal channel would be more effective at reducing outliers than rotations between two normal channels. We validate this intuition with a simple experiment: for a linear layer with many outliers, optimizing only the top 10% channel pairs with *largest magnitude difference* is almost as effective at reducing quantization-induced output error as optimizing all pairs (Figure 2). This creates an opportunity for designing parameter-efficient and potentially inference-efficient rotations for addressing the outlier issue: by retaining only the rotations between channel pairs that have large magnitude differences, we can maintain the effectiveness of a full $n \times n$ orthogonal matrix.

## 4 METHOD

In this section, we introduce ParoQuant, a weight-only quantization method that applies optimized parameter- and inference-efficient rotations to effectively reduce quantization error. We start with the design of our *scaled pairwise rotation* transform. Then, we introduce the algorithm to optimize the transform and fine-tune the quantized models. Finally, we provide an efficient kernel that enables extremely fast inference. Our focus in this paper is on *linear quantization* as it is more efficient than vector quantization and is better supported by existing inference frameworks (Lin et al., 2024b; Zheng et al., 2024; Kwon et al., 2023), though the same method can be extended to vector quantization.

### 4.1 SCALED PAIRWISE ROTATION

We follow a three-step process to design our scaled pairwise rotation transform. First, we avoid direct matrix multiplications by replacing orthogonal matrices with decomposed *Givens rotations*. Next, we remove dependencies among these rotations to enable parallel execution on GPUs, resulting in *independent rotation*. Finally, because a single independent rotation is not effective enough to fit complex weight distributions, we sequentially apply a *series of independent rotations* combined with *channel-wise scaling* to improve the fitting capability.

#### 4.1.1 GIVENS ROTATION

Based on the observation in Section 3 that most parameters in an orthogonal matrix are redundant, we can select a small set of channel pairs $\mathcal{P} = \{(i_1, j_1), \ldots, (i_m, j_m)\}$ and sequentially rotate each pair in $\mathcal{P}$ instead of performing full matrix multiplication. Formally, given $\mathcal{P}$, a set of rotation angles $\Theta = \{\theta_1, \ldots, \theta_m\}$, and the weight matrix $\mathbf{W}$, the transformed weight is

$$\mathbf{W}^{(m)} = G(i_m, j_m, \theta_m) \, G(i_{m-1}, j_{m-1}, \theta_{m-1}) \cdots G(i_1, j_1, \theta_1) \, \mathbf{W}, \tag{3}$$

where $G(i_k, j_k, \theta_k)$ is a Givens rotation that rotates two rows of the matrix while keeping others intact. This operation can be applied in place with just a few vectorized multiply-and-add instructions:

$$\begin{aligned} \mathbf{W}^{(k)}[i, :] &= \cos \theta_k \cdot \mathbf{W}^{(k-1)}[i, :] - \sin \theta_k \cdot \mathbf{W}^{(k-1)}[j, :], \\ \mathbf{W}^{(k)}[j, :] &= \sin \theta_k \cdot \mathbf{W}^{(k-1)}[i, :] + \cos \theta_k \cdot \mathbf{W}^{(k-1)}[j, :]. \end{aligned} \tag{4}$$

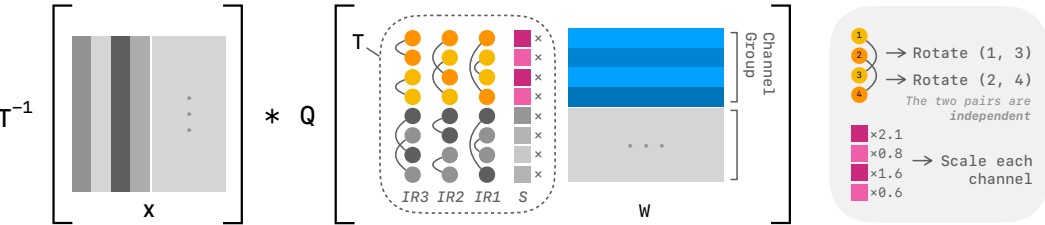

Figure 3: Overview of scaled pairwise rotation ($T$). The channel dimension is divided into fixed-size groups (the group size is 4 in the figure). Each group of the weights (**W**) is transformed by channel-wise scaling (S), followed by a series of independent rotations (IR). Each independent rotation consists of pairwise rotations that are mutually independent (*i.e.*, non-overlapping). Quantization ($Q$) is applied after the transform using a group size equal to the channel group size. The inverse transform ($T^{-1}$) is applied to the activations (**X**).

The actual computation during inference is applying the inverse of the Givens rotation sequence in Equation (3) to the activations **X**. The inverse can be conveniently obtained by reversing the sequence and replacing each $\theta_k$ with $-\theta_k$:

$$\mathbf{X}^{(m)} = \mathbf{X}\, G(i_1, j_1, -\theta_1)\, G(i_2, j_2, -\theta_2) \cdots G(i_m, j_m, -\theta_m), \tag{5}$$

which can also be computed efficiently, similar to Equation (4).

### 4.1.2 INDEPENDENT ROTATION

Givens rotations eliminate the need for matrix multiplications, but they remain inefficient due to potential dependencies. Such dependencies arise when a channel rotates with more than one other channel. In these cases, Givens rotations are not commutative, and the order in which they are applied matters. As a result, dependent Givens rotations must be computed sequentially and cannot fully exploit the GPU's massive parallelism, leading to significant latency. To address this issue, we require the pairs within $\mathcal{P}$ to be mutually *independent*, *i.e.*, each channel may appear in only one pair. Under this constraint, it follows directly from Equation (4) that the computation for each pair is completely independent and does not interfere with any other pair. Consequently, all Givens rotations for $\mathcal{P}$ are *fully parallelizable*. The same conclusion applies to Equation (5).

In addition to computational efficiency, another benefit of independent rotations is their intrinsic compatibility with block-wise quantization. In block-wise quantization, an outlier channel within a group can only impact the quantization accuracy of other channels within the same group. Naturally, we can exploit the isolation between groups by applying a separate independent rotation for each group. This enables fine-grained pair selections specific to each group and allows a higher degree of parallelism (see Section 4.3).

We formulate independent pairs and independent rotation as follows:

**Definition 1** (Independent Pairs). Consider a set of pairs $\mathcal{P} = \{(i_1, j_1), \ldots, (i_n, j_n)\}$, and let each pair $(i_k, j_k)$ be represented as a set $P_k = \{i_k, j_k\}$. $\mathcal{P}$ is a set of *independent pairs* if and only if:

$$\forall P_k, P_l \in \{P_1, \ldots, P_n\} \text{ where } k \neq l, \quad P_k \cap P_l = \emptyset. \tag{6}$$

**Definition 2** (Independent Rotation). Consider the product of a set of Givens rotations on pairs $\mathcal{P} = \{(i_1, j_1), \ldots, (i_n, j_n)\}$ with the corresponding angles $\Theta = \{\theta_1, \ldots, \theta_n\}$:

$$R(\mathcal{P}, \Theta) = \prod_{k=1}^{n} G(i_k, j_k, \theta_k), \tag{7}$$

we say $R(\mathcal{P}, \Theta)$ is an *independent rotation* if and only if $\mathcal{P}$ is a set of independent pairs.

### 4.1.3 SERIES OF INDEPENDENT ROTATIONS

With dependencies eliminated, independent rotations can be applied online during inference with very small overhead. However, an independent rotation of dimension $n$ can accommodate only $\frac{n}{2}$

independent pairs, which correspond to $\frac{n}{2}$ tunable parameters. This is only a fraction $\frac{1}{n-1}$ of the $\frac{1}{2}n(n-1)$ parameters in a full orthogonal matrix and thus severely limits its fitting capability. To overcome this limitation, instead of using only one independent rotation, we sequentially apply a small number (*e.g.*, 8) of them to improve the expressiveness of the transform. Multiple rotations can be fused into a single kernel with a one-time memory load with minimal overhead (see Section 4.3).

Algorithm A1 describes how ParoQuant selects pairs for a series of independent rotations. For each rotation, we randomly select available pairs while ensuring the rotation remains independent. To enable more diverse combinations of channel pairs across different independent rotations, we skip pairs that have already been selected in previous rotations. This constraint may result in an insufficient number of pairs for some rotations, but the impact is negligible in practice.

### 4.1.4 COMBINING CHANNEL-WISE SCALING

On top of a series of independent rotations, we apply channel-wise scaling to further reduce quantization error. Because independent rotations act on only a limited number of pairs ($\mathcal{O}(n)$ *vs.* $\mathcal{O}(n^2)$ for a full rotation), the ability of channel-wise scaling to directly even out the magnitudes across the *entire* matrix is crucial for our transform to match the expressiveness of full rotations. It is also more straightforward to suppress *isolated* outliers with channel-wise scaling than with Givens rotations.

After combining independent rotations with channel-wise scaling, the final transform (*i.e.*, scaled pairwise rotation) applied to the weights before quantization is:

$$T_{\mathcal{P},\Theta,\boldsymbol{\alpha}}(\mathbf{W}) = \left(\prod_{t=1}^{K} R(\mathcal{P}_t, \Theta_t)\right) \cdot \mathrm{diag}(\boldsymbol{\alpha}) \cdot \mathbf{W}, \tag{8}$$

where $K$ is the number of rotations, $\mathcal{P} = \{\mathcal{P}_1, \ldots, \mathcal{P}_K\}$ and $\Theta = \{\Theta_1, \ldots, \Theta_K\}$ are the corresponding sets of rotation pairs and angles, $R(\mathcal{P}_t, \Theta_t)$ is the $t$-th independent rotation, and $\boldsymbol{\alpha}$ is the set of per-channel scaling factors. Integrating channel-wise scaling is efficient, as it can be fused into the rotation kernel at minimal cost. We refer the readers to Section 5.3 and Section A.2 for the effectiveness of independent rotations and channel-wise scaling.

### 4.2 LAYER-WISE OPTIMIZATION

To optimize the scaled pairwise rotation in Equation (8), we adopt a layer-wise optimization scheme to minimize the output loss of each layer. Specifically, for a decoder layer $D$, we minimize

$$\mathcal{L}(Q) = \|Q(D)(\mathbf{X}') - D(\mathbf{X})\|, \tag{9}$$

where $Q(D)$ is the decoder $D$ with every linear layer quantized after applying the scaled pairwise rotation, $\mathbf{X}$ is the input to $D$ of the original model, and $\mathbf{X}'$ is the output of the *already quantized* preceding decoder layers. By optimizing with the new output computed from $\mathbf{X}'$ instead of from $\mathbf{X}$, the subsequent layers can compensate for quantization errors introduced by earlier layers, thereby improving end-to-end accuracy.

For each layer, we optimize the quantized model in two stages. In the first stage, we optimize rotations and channel-wise scaling. After this stage, most outliers in the weight matrices are suppressed, and the weights are more quantization-friendly. However, some isolated outliers may still remain, as rotations and scaling cannot eliminate them completely. Therefore, in the second stage, we adopt a QAT-like approach similar to EfficientQAT (Chen et al., 2025) to fine-tune the weights and the linear quantization parameters $s$ and $z$ in Equation (1), thereby further reducing the error introduced by the RTN algorithm. The pseudocode for the optimization algorithm is available in Section A.1.

### 4.3 CO-DESIGNING EFFICIENT TRANSFORM KERNEL

To enable fast inference, we implement the scaled pairwise rotation transform as a single fused CUDA kernel. Thanks to the transform's independence at both the group and pair levels, the computation is fully parallelized at three levels: (1) *token*: we parallelize across the token dimension of the activation tensor; (2) *channel group*: we assign different CUDA blocks to different groups along the channel dimension; (3) *pair*: each rotation pair is processed by a separate CUDA thread.

This fine-grained parallelism across groups and pairs offers several advantages. First, dividing the channel dimension into groups reduces the memory load required for each thread block. Because the group size (*e.g.*, 128) is relatively small, the activation tensor fits into the on-chip shared memory, and the rotation parameters (*i.e.*, pair indices and angles) fit into registers. This significantly reduces the latency of subsequent memory access. As a result, multiple independent rotations can then be fused efficiently, since the activation and all parameters are already loaded into low-latency memory. Second, group-wise parallelism increases the occupancy of the GPU's compute units, particularly when the channel dimension is very large. From Figure 4, the speedup of our transform (with 8 independent rotations) over the fast Hadamard transform (Dao, 2024)

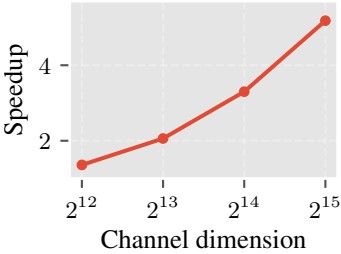

Figure 4: Speedup of scaled pairwise rotation over the Hadamard transform on an RTX A6000.

increases with the channel dimension, because the Hadamard transform has inherent dependencies across all channels. Third, pair-level independence within each rotation allows synchronization-free execution across all CUDA threads within a thread block, further improving hardware utilization.

## 5 EVALUATION

**Models and Tasks.** We apply ParoQuant on LLaMA-2 (7B) (Touvron et al., 2023), LLaMA-3 (8B, 70B) & LLaMA-3.1 Instruct (8B) (Grattafiori et al., 2024), DeepSeek-R1-distilled LLaMA-3.1 (8B) (Guo et al., 2025), and Qwen3 (1.7B, 4B, 8B, 14B) (Yang et al., 2025) pre-trained models. We evaluate the quantization quality with three types of evaluation: (1) *Perplexity* on WikiText2 (Merity et al., 2017) and C4 (Dodge et al., 2021); (2) *Reasoning accuracy* on MMLU-Pro (Wang et al., 2024), GPQA Diamond (Rein et al., 2024), AIME-24, and AIME-25 (MAA, 2025); (3) *Non-reasoning accuracy* on BoolQ (Clark et al., 2019), ARC-Challenge, ARC-Easy (Clark et al., 2018), and HellaSwag (Zellers et al., 2019).

**Implementation.** We focus on 4-bit weight-only linear quantization with a group size of 128. Linear quantization is more efficient and is widely supported by existing frameworks. The choice of 4 bits and a 128 group size offers the optimal trade-off between accuracy and bit width for linear quantization (Dettmers & Zettlemoyer, 2023). We apply 8 independent rotations on each 128-channel group, with each rotation consisting of up to 64 pairs. Each layer is optimized for 10 epochs at each stage using AdamW (Loshchilov & Hutter, 2019) with a fixed set of hyperparameters for all experiments, except for the 70B model, where we adjust the batch size to accommodate memory constraints. To reduce the risk of overfitting to one dataset, we use a training set of 2048 samples drawn evenly from WikiText2, C4, and RedPajama (Weber et al., 2024), and select the best parameters using 64 validation samples from Pile (Gao et al., 2020). More details are provided in Section A.4.

**Baselines.** We compare the accuracy and efficiency of ParoQuant with three weight-only PTQ baselines. AWQ (Lin et al., 2024b) optimizes channel-wise scaling with grid search and is the most used 4-bit weight-only quantization method. EfficientQAT (Chen et al., 2025) achieves state-of-the-art linear quantization accuracy with layer-wise fine-tuning of weights and quantization parameters[*]. QTIP (Tseng et al., 2024b) is the state-of-the-art vector quantization method utilizing randomized Hadamard transform and an advanced trellis quantization algorithm. In addition, we include the perplexity results of QuIP# (Tseng et al., 2024a), a vector-quantization predecessor of QTIP that also adopts the Hadamard transform, and two weight-activation linear quantization methods, OmniQuant (Shao et al., 2024) and SpinQuant (Liu et al., 2025b), which are also applicable for weight-only quantization. We apply block-wise quantization with a group size of 128 on all linear quantization methods and the corresponding default settings on vector quantization methods.

### 5.1 ACCURACY RESULTS

**Perplexity.** Table 1 shows the perplexity results of 4-bit quantized models ranging in size from 1.7B to 70B. Among linear quantization methods, ParoQuant achieves state-of-the-art quantization

---

[*]We only apply the "Block-AP" stage of EfficientQAT, as its "E2E-QP" stage involves supervised fine-tuning, which is out of the scope of PTQ.

accuracy across all sizes, particularly on challenging cases like LLaMA-3 and smaller models under 4B. It also delivers strong performance compared with rotation-based methods including QuIP#, QTIP, and SpinQuant. It outperforms QuIP# and matches QTIP on all models, despite the inherently larger error of linear quantization, highlighting the superior effectiveness of our proposed transform over the Hadamard transform (see Section A.2 for detailed analysis). Moreover, ParoQuant provides a decent speedup over these two methods. This underscores the efficiency of our proposed transform.

| Method | Type | WikiText2 | | | | | | | C4 | | | | | | | Speedup |
|---|---|---|---|---|---|---|---|---|---|---|---|---|---|---|---|---|
| | | L3-8 | L3-70 | L2-7 | Q3-1.7 | Q3-4 | Q3-8 | Q3-14 | L3-8 | L3-70 | L2-7 | Q3-1.7 | Q3-4 | Q3-8 | Q3-14 | |
| FP16 | – | 5.54 | 2.56 | 5.12 | 8.32 | 7.01 | 6.24 | 5.70 | 7.10 | 5.78 | 6.63 | 8.62 | 7.61 | 6.97 | 6.54 | 1.0× |
| QuIP# | vector | 5.81 | 2.99 | 5.19 | – | – | – | – | 7.32 | 5.96 | 6.75 | – | – | – | – | 1.9× |
| QTIP | vector | 5.69 | 2.75 | 5.17 | 8.46 | 7.09 | 6.28 | 5.75 | 7.22 | 5.83 | 6.69 | 8.73 | 7.68 | 7.02 | 6.57 | 1.7× |
| AWQ | linear | 5.92 | 2.96 | 5.23 | 8.80 | 7.36 | 6.45 | 5.85 | 7.42 | 5.91 | 6.80 | 9.01 | 7.89 | 7.14 | 6.65 | 2.4× |
| OmniQ | linear | – | – | 5.23 | – | – | – | – | – | – | 6.80 | – | – | – | – | 2.4×[†] |
| SpinQ | linear | 5.83 | – | 5.21 | – | – | – | – | 7.41 | – | 6.86 | – | – | – | – | 2.4×[†] |
| E-QAT | linear | 5.87 | 3.33 | 5.22 | 8.60 | 7.19 | 6.37 | 5.82 | 7.36 | 6.72 | 6.76 | 8.84 | 7.77 | 7.08 | 6.63 | 2.4×[†] |
| ParoQ | linear | **5.73** | **2.82** | **5.17** | **8.44** | **7.10** | **6.29** | **5.75** | **7.27** | **5.86** | **6.73** | **8.74** | **7.70** | **7.04** | **6.59** | 2.2× |

[†] Uses results of AWQ as a reference, as the method does not incur significant overhead from the transform.

Table 1: Perplexity (↓) results of 4-bit models. The context length is 8192 for LLaMA-3 and Qwen3 (base models), and 4096 for LLaMA-2. The best results among linear quantization methods are in **bold**. Speedup over FP16 models is reported as the geometric mean across Q3-1.7, Q3-4, L3-8, Q3-14, measured on an RTX A6000 with a batch size of 1 during decoding.

**Reasoning Tasks.** Table 2 shows the accuracy results of four reasoning benchmarks: MMLU-Pro (12k samples), GPQA Diamond (198 samples), AIME-24 (30 samples), and AIME-25 (30 samples). On the larger MMLU-Pro benchmark, ParoQuant consistently outperforms all linear quantization baselines and matches the accuracy of QTIP. While results on the smaller GPQA and AIME benchmarks exhibit more randomness due to the limited number of samples, ParoQuant still outperforms the baselines in most cases. Overall, ParoQuant causes only an average **0.9%** accuracy degradation and achieves **6.3%**, **2.4%**, and **0.9%** improvements over EfficientQAT, AWQ, and QTIP, respectively. This demonstrates ParoQuant's superior quantization accuracy in long generation.

| Method | Type | R1-Distill-Llama-8B | | | | Qwen3-4B | | | | Qwen3-8B | | | | Qwen3-14B | | | | Avg. |
|---|---|---|---|---|---|---|---|---|---|---|---|---|---|---|---|---|---|---|
| | | MMLU | GPQA | AIME 24 | AIME 25 | MMLU | GPQA | AIME 24 | AIME 25 | MMLU | GPQA | AIME 24 | AIME 25 | MMLU | GPQA | AIME 24 | AIME 25 | |
| FP16 | – | 58.8 | 46.6 | 42.2 | 32.2 | 71.0 | 50.0 | 75.6 | 62.2 | 74.6 | 60.3 | 75.6 | 72.2 | 78.1 | 62.5 | 73.3 | 68.9 | 62.8 |
| QTIP | vector | 57.4 | 43.4 | 37.8 | 30.1 | 69.7 | 55.2 | 67.8 | 58.9 | 74.0 | 59.3 | 72.2 | 63.3 | 77.9 | 64.0 | 76.7 | 69.0 | 61.0 |
| AWQ | linear | 56.0 | 44.1 | 34.4 | 26.7 | 68.2 | 52.2 | 62.2 | 53.3 | 73.5 | **60.2** | 72.2 | 61.1 | 77.2 | 62.0 | **80.0** | **68.9** | 59.5 |
| E-QAT | linear | 55.6 | 44.6 | 32.2 | 21.1 | 67.7 | 51.0 | 53.3 | 46.7 | 73.4 | 56.1 | 68.9 | 51.1 | 76.5 | 58.4 | 71.1 | 61.1 | 55.6 |
| ParoQ | linear | **57.1** | **47.5** | **36.6** | **31.1** | **70.1** | **53.7** | **73.3** | **63.3** | **74.1** | 57.7 | **75.6** | **63.3** | **77.5** | **63.5** | 77.8 | 67.8 | **61.9** |

Table 2: Zero-shot accuracy (↑) on reasoning tasks. Best linear quantization results are in **bold**.

**Non-Reasoning Tasks.** Table 3 shows the zero-shot accuracy on commonsense benchmarks with thinking mode disabled. ParoQuant maintains near-lossless performance, outperforming AWQ, EfficientQAT, and QTIP by **0.9%**, **0.7%**, and **0.2%**, respectively. The accuracy gap is smaller than in reasoning tasks because these benchmarks evaluate only a few generated tokens, so error accumulation is minimal.

## 5.2 Efficiency Results

Table 4 shows the decoding throughput on an RTX A6000. To ensure a fair comparison, we implement all methods on top of the Transformers library (Wolf et al., 2020), modifying only the weight transform and dequantization code (details and more results are in Section A.5). ParoQuant is only about 10% slower than AWQ while providing a significant accuracy improvement, and it matches the accuracy of QTIP while being 15%-30% faster. For the training efficiency, see Section A.6 for more details.

| Method | Type | LLaMA-3.1-8B-Instruct | | | | Qwen3-4B | | | | Qwen3-8B | | | | Qwen3-14B | | | | Avg. |
|---|---|---|---|---|---|---|---|---|---|---|---|---|---|---|---|---|---|---|
| | | BoolQ | ARC-C | ARC-E | HSwag | BoolQ | ARC-C | ARC-E | HSwag | BoolQ | ARC-C | ARC-E | HSwag | BoolQ | ARC-C | ARC-E | HSwag | |
| FP16 | – | 84.1 | 51.7 | 81.8 | 59.1 | 85.1 | 50.8 | 80.5 | 52.3 | 86.6 | 55.8 | 83.5 | 57.1 | 89.4 | 58.6 | 84.2 | 60.9 | 70.1 |
| QTIP | vector | 84.3 | 51.8 | 81.6 | 58.9 | 85.0 | 50.0 | 79.8 | 51.8 | 86.9 | 54.9 | 82.8 | 57.0 | 89.2 | 57.6 | 83.5 | 60.8 | 69.7 |
| AWQ | linear | 83.5 | 51.7 | 80.6 | 58.4 | 85.0 | 47.4 | 77.9 | 51.3 | 86.2 | 53.8 | 82.2 | 56.2 | **89.1** | 57.9 | 83.2 | 60.3 | 69.0 |
| E-QAT | linear | 83.5 | 51.9 | 80.9 | 58.4 | 84.5 | 48.3 | 79.7 | 51.1 | 86.1 | 53.6 | 81.7 | 56.1 | 89.0 | **58.5** | 84.0 | 60.4 | 69.2 |
| PAROQ | linear | **83.9** | **52.1** | **82.2** | **58.7** | **85.3** | 49.7 | **80.7** | 51.8 | **87.0** | **55.3** | **83.3** | **56.8** | **89.1** | 57.2 | **84.3** | **60.7** | **69.9** |

Table 3: Zero-shot accuracy (↑) on non-reasoning tasks. Best linear quantization results are in **bold**.

| Method | Qwen3-1.7B | | Qwen3-4B | | LLaMA-3-8B | | Qwen3-14B | |
|---|---|---|---|---|---|---|---|---|
| | Throughput | W2 PPL | Throughput | W2 PPL | Throughput | W2 PPL | Throughput | W2 PPL |
| FP16 | 170 (1.0×) | 8.32 | 78 (1.0×) | 7.01 | 45 (1.0×) | 5.54 | 25 (1.0×) | 5.70 |
| AWQ | 320 (1.9×) | 8.80 | 176 (2.3×) | 7.36 | 120 (2.7×) | 5.92 | 70 (2.8×) | 5.85 |
| QTIP | 209 (1.2×) | 8.46 | 117 (1.5×) | 7.09 | 95 (2.1×) | 5.69 | 55 (2.2×) | 5.75 |
| PAROQ | 278 (1.6×) | 8.44 | 160 (2.1×) | 7.10 | 112 (2.5×) | 5.73 | 65 (2.6×) | 5.75 |

Table 4: Decoding (with batch size of 1) throughput (tokens/s).

## 5.3 ABLATION STUDY

Table 5 shows the effectiveness of each component of ParoQuant. The effects of channel-wise scaling and independent rotations are distinct, and combining both of them yields better quantization accuracy than applying either one alone. Fine-tuning weights and quantization parameters in the second optimization stage further improves the accuracy compared with directly applying RTN. For a more detailed comparison of the transforms, see Section A.2.

Table 6 shows the effects of the calibration set, calibration size, and number of independent rotations on end-to-end quantization accuracy. ParoQuant achieves surprisingly strong performance with as few as 128 training samples. Moreover, accuracy improves as the number of rotations increases up to 8, indicating improved fitting capability. We also optimize the model with 2048 calibration samples from RedPajama alone, and the results are slightly worse than with the mixed dataset. This shows that using a more diverse training set improves the generalization ability of the models.

| | Transform | C4 (↓) |
|---|---|---|
| | None | 7.56 |
| | S | 7.40 |
| w/o Stage 2 | 8 IR | 7.50 |
| | 8 IR + S | 7.35 |
| | None | 7.42 |
| | S | 7.41 |
| w/ Stage 2 | 8 IR | 7.40 |
| | 8 IR + S | 7.27 |

Table 5: Ablations on transforms and optimization stages with LLaMA-3-8B (S: channel-wise scaling, IR: independent rotation).

| # Samples | # IR | C4 (↓) | MMLU (↑) |
|---|---|---|---|
| 128 | 8 | 7.30 | 69.5 |
| 512 | 8 | 7.27 | 69.7 |
| 2048 | 0 | 7.41 | 69.6 |
| | 2 | 7.28 | 69.4 |
| | 4 | 7.27 | 69.4 |
| | 8 | 7.27 | 70.1 |
| 2048 (RedPajama) | 8 | 7.27 | 69.5 |

Table 6: Ablations on training samples and number of rotations (IR) with LLaMA-3-8B (C4 perplexity) and Qwen3-4B (MMLU-Pro accuracy).

## 6 CONCLUSION

In this paper, we proposed ParoQuant, an efficient PTQ method that achieves state-of-the-art quantization accuracy with minimal overhead. Based on the insight that a sparsely parameterized rotation can effectively suppress weight outliers, we designed scaled pairwise rotation, which combines hardware-friendly independent Givens rotations with channel-wise scaling. ParoQuant matches the

accuracy of the best existing quantization methods while running much faster, and it consistently outperforms prior efficient quantization methods, especially on reasoning tasks where quantization errors accumulate over long chains of thought. We hope that our method will inspire future research on high-fidelity, low-overhead quantization techniques for next-generation reasoning LLMs.

## ACKNOWLEDGMENT

We sincerely thank Zihan Zhang for assistance with the reasoning task evaluation and Shang Yang for valuable feedback on earlier drafts of this work.

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

## A.1 CORE ALGORITHMS

---

**Algorithm A1:** Selection of Independent Channel Pairs

---

**Input:** $\mathbf{W} \in \mathbb{R}^{g \times D}$: a subgroup of the weight sliced along channel dimension, where $g$ is the group size; $K$: number of independent rotations; $N$: number of pairs per rotation.
**Output:** $\mathcal{P}_1, \ldots, \mathcal{P}_K$: lists of selected pairs for each rotation.
$\mathcal{P} \leftarrow \{(i,j) \mid 1 \le i < j \le g\}$
$\mathcal{P}_{\text{shuffled}} \leftarrow \text{Shuffle}(\mathcal{P})$
// Matrix to track available pairs across all rotations
Initialize $\mathbf{A} \in \mathbb{R}^{g \times g}$ where $\mathbf{A}_{ij} \leftarrow 1, \forall i,j \in \{1,...,g\}, i \ne j$ and $\mathbf{A}_{ii} \leftarrow 0, \forall i \in \{1,...,g\}$
$\mathcal{P}_1, \ldots, \mathcal{P}_K \leftarrow [\,]$
**for** $r \leftarrow 1$ **to** $K$ **do**
  $\quad \mathbf{A}_{\text{rot}} \leftarrow \text{Copy}(\mathbf{A})$ \qquad // Tracks available channels within this rotation
  $\quad$ **foreach** pair $(i,j) \in \mathcal{P}_{\text{shuffled}}$ **do**
    $\qquad$ **if** $|\mathcal{P}_r| = N$ **then break**
    $\qquad$ **if** $\mathbf{A}_{\text{rot}}[i,j] = 0$ **then continue**
    $\qquad$ Append $(i,j)$ to $\mathcal{P}_r$ \qquad // Select the next available pairs
    $\qquad \mathbf{A}_{\text{rot}}[i,:] \leftarrow 0; \mathbf{A}_{\text{rot}}[:,i] \leftarrow 0; \mathbf{A}_{\text{rot}}[j,:] \leftarrow 0; \mathbf{A}_{\text{rot}}[:,j] \leftarrow 0$ \quad // Block channels
    $\qquad \mathbf{A}[i,j] \leftarrow 0; \mathbf{A}[j,i] \leftarrow 0$ \qquad\qquad // Block pair

**return** $\mathcal{P}_1, \ldots, \mathcal{P}_K$

---

---

**Algorithm A2:** Layer-Wise Optimization

---

**Input:** $\mathcal{L}$: decoder layers of the model; $g$: group size; $K$: number of rotations; $N$: number of pairs per rotation; $\mathcal{D}$: calibration dataset.
**Output:** $\mathcal{L}'$: decoder layers containing optimized scaling, rotations, and quantizers.
$\mathcal{X} \leftarrow \text{tokenize } \mathcal{D}$ \qquad // $\mathcal{X}$ is the original input to a layer
$\mathcal{X}' \leftarrow \mathcal{X}$ \qquad // $\mathcal{X}'$ is the input to a layer after quantizing preceding layers
$\mathcal{L}' \leftarrow [\,]$
**foreach** layer $l \in \mathcal{L}$ **do**
  $\quad \mathcal{Y} \leftarrow l(\mathcal{X})$ \qquad // Output of original layer as labels
  $\quad l' \leftarrow \text{Copy}(l)$
  $\quad$ **foreach** $linear \in l'$ **do**
    $\qquad \mathbf{W}_1, \ldots, \mathbf{W}_n \leftarrow \text{Partition } linear$ with group size $g$ at channel dimension
    $\qquad$ **for** $i \leftarrow 1$ **to** $n$ **do**
      $\qquad\quad \mathcal{P}_i \leftarrow \text{SelectPairs}(\mathbf{W}_i, K, N)$ \qquad // Algorithm A1
      $\qquad\quad \boldsymbol{\theta}_i \leftarrow \mathbf{0}_{K \times N}, \boldsymbol{\alpha}_i \leftarrow \mathbf{1}_g$ \qquad // Angles and channel-wise scaling
      $\qquad\quad \mathbf{s}_i, \mathbf{z}_i \leftarrow \text{Initialize quantizer with } \mathbf{W}_i$ \qquad // Equation (1)
      $\qquad\quad$ Insert scaling $\boldsymbol{\alpha}_i$, rotations $(\mathcal{P}_i, \boldsymbol{\theta}_i)$, and quantizer $(\mathbf{s}_i, \mathbf{z}_i)$ into $l'$
  $\quad l'' \leftarrow \text{Optimize all } \boldsymbol{\theta}_i \text{ and } \boldsymbol{\alpha}_i \text{ to minimize } \|\mathcal{Y} - l'(\mathcal{X}')\|$ \qquad // Stage 1
  $\quad l''' \leftarrow \text{Optimize all } \mathbf{s}_i, \mathbf{z}_i, \text{ and weights to minimize } \|\mathcal{Y} - l''(\mathcal{X}')\|$ \qquad // Stage 2
  $\quad$ Append $l'''$ to $\mathcal{L}'$
  $\quad \mathcal{X}' \leftarrow l'''(\mathcal{X})$ \qquad // Quantized layers' output as next layer's input
  $\quad \mathcal{X} \leftarrow \mathcal{Y}$ \qquad // Pass down original layer's output
**return** $\mathcal{L}'$

---

## A.2 EFFECTIVENESS ANALYSIS

To study the effectiveness of different transforms at minimizing quantization-induced output error ($\|\mathbf{X}Q(\mathbf{W}) - \mathbf{X}\mathbf{W}\|$), we optimize each linear layer individually with each transform applied to the input dimension of its weight $\mathbf{W}$ and obtain the loss curves of the optimization process. We compare the effectiveness of five transforms:

- **Channel-wise scaling**: We optimize per-channel scaling factors $\boldsymbol{\alpha}$ for the weight $\mathbf{W}$, *i.e.*, $\hat{\mathbf{W}} = \mathrm{diag}(\boldsymbol{\alpha}) \cdot \mathbf{W}$.
- **Full rotation**: We construct an orthogonal matrix $R$ from an upper triangular matrix $U$ by $R = \exp(U - U^T)$ (exp is the matrix exponential), and optimize the elements of $U$.
- **Random Hadamard transform**: We sample the average output error after applying a random Hadamard transform generated using one of 100 seeds.
- **Independent rotation**: We select 8 independent rotations for each 128-channel group using Algorithm A1.
- **Scaled pairwise rotation** (independent rotation + channel-wise scaling): We select 8 independent rotations for each 128-channel group using Algorithm A1.

To optimize the transforms, we use AdamW with a learning rate of 0.001 for the full rotation and 0.01 for other transforms. We use 128 samples from Pile and optimize for 200 steps.

The results for the first, middle, and last layer of LLaMA-3-8B are listed in Figure A1. `mlp.down_proj` results are not included because the input dimension is too large for the full rotation. From the results, independent rotations can lower quantization error better than channel-wise scaling in layers that possess many outliers (*e.g.*, `q_proj` and `k_proj`), and generally outperform random Hadamard transform with much lower overhead. When combined with channel-wise scaling, independent rotations can almost match the effectiveness of a full rotation in layers with many outliers. This proves the superior expressiveness of our proposed scaled pairwise rotation transform.

Figure A1: Loss curves from optimizing transforms to minimize quantization-induced output error of linear layers in LLaMA-3-8B.

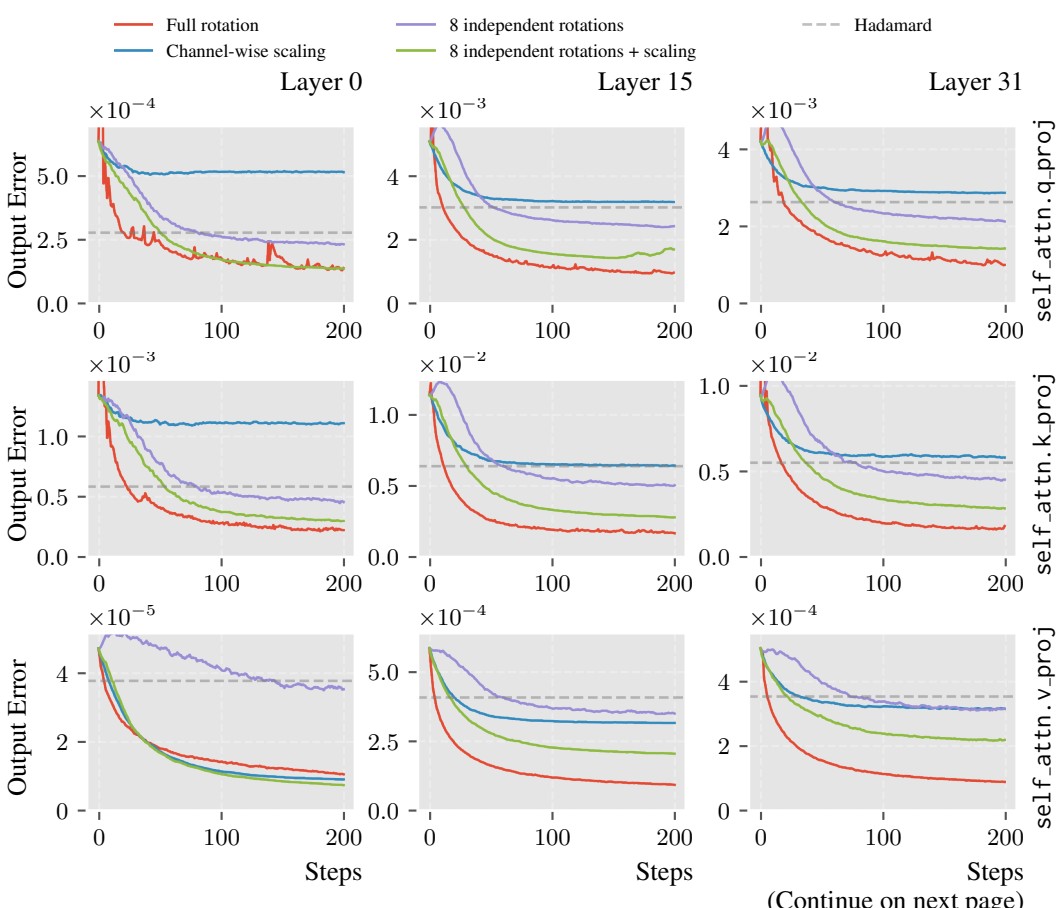

(Continue on next page)

Figure A1: Loss curves from optimizing transforms to minimize quantization-induced output error of linear layers in LLaMA-3-8B (continued).

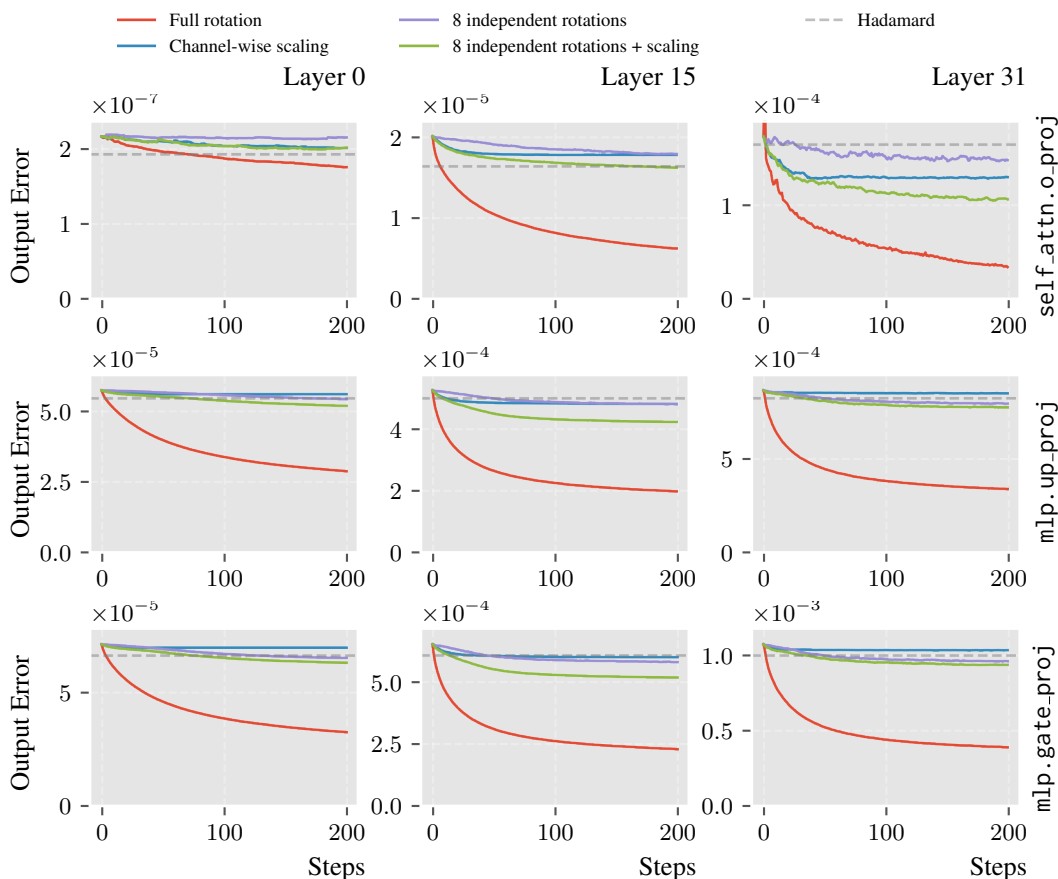

## A.3 EXTENSION TO WEIGHT-ACTIVATION QUANTIZATION

In this section, we extend ParoQuant to 4-bit weight-activation (W4A4) quantization. We did not tune ParoQuant specifically for W4A4, and thus the results may not be optimal; however, we share these preliminary findings to facilitate further research.

**Weight-Activation Quantization.** Similar to Equation (2), a linear layer $\mathbf{Y} = \mathbf{X}\mathbf{W} + \mathbf{b}$ is reformulated and quantized as

$$\mathbf{Y}' = Q(\mathbf{X}\mathbf{T}^{-1}) \cdot Q(\mathbf{T}\mathbf{W}) + \mathbf{b}. \tag{10}$$

By quantizing activations after the inverse transform $T^{-1}$, the matrix multiplication is performed entirely in low precision.

**Per-Linear Optimization with GPTQ.** We discover that per-decoder optimization often fails to converge under 4-bit activation quantization, so we employ a per-*linear* strategy with error compensation. Specifically, modules within a decoder are optimized sequentially using the quantized outputs of all preceding modules as calibration inputs, enabling downstream layers to adapt to and compensate for accumulated quantization error. In addition, naively optimizing the weights during the fine-tuning stage used in the W4A16 setting is unstable, so we instead adopt GPTQ (Frantar et al., 2023) to recover the precision of the transformed weights $T(\mathbf{W})$ instead of fine-tuning.

**Evaluation.** We evaluate ParoQuant under the W4A4 setting using the same evaluation setup described in Section A.7. For INT4 quantization, we compare ParoQuant with two state-of-the-art methods SpinQuant and FlatQuant (Sun et al., 2025). We also apply ParoQuant to the MXFP4 format, which is a hardware-native microscaling format, and we compare it to MR-GPTQ (Egiazarian et al., 2025). All methods employ GPTQ for weight quantization.

| Method | W2 ($\downarrow$) | C4 ($\downarrow$) | Non-reasoning ($\uparrow$) | Reasoning ($\uparrow$) |
|---|---|---|---|---|
| FP16 | 6.24 | 6.97 | 70.8 | 70.7 |
| SPINQUANT | 7.13 | 7.89 | 66.6 | 61.7 |
| FLATQUANT | 6.51 | 7.25 | 69.4 | 63.9 |
| PAROQUANT | 6.62 | 7.36 | 67.7 | 63.6 |

Table A2: INT4 W4A4 quantization results.

| Method | W2 ($\downarrow$) | C4 ($\downarrow$) | Non-reasoning ($\uparrow$) | Reasoning ($\uparrow$) |
|---|---|---|---|---|
| FP16 | 6.24 | 6.97 | 70.8 | 70.7 |
| MR-GPTQ | 6.99 | 7.71 | 68.4 | 62.5 |
| PAROQUANT | 6.75 | 7.44 | 69.6 | 61.4 |

Table A3: MXFP4 W4A4 quantization results.

For INT4, ParoQuant outperforms SpinQuant by a large margin and is comparable with FlatQuant, both of which are specifically optimized for weight-activation quantization. For MXFP4, ParoQuant also outperforms the state-of-the-art MR-GPTQ, which is designed for microscaling formats. These preliminary results demonstrate the effectiveness of ParoQuant's scaled pairwise rotation and indicate strong potential for further improvements in weight-activation quantization performance.

## A.4 IMPLEMENTATION DETAILS

**Quantization and Transform.** We apply block-wise linear quantization with a group size of 128. We apply channel-wise scaling and 8 independent rotations on each 128-channel group of the weights, with each rotation consisting of up to 64 pairs. The independent rotations are applied sequentially after channel-wise scaling.

**Libraries.** Our implementation is built with PyTorch 2.8.0 (Paszke et al., 2019), Transformers 4.55.2 (Wolf et al., 2020), and Datasets 3.6.0 (Lhoest et al., 2021).

**Optimization.** We optimize all ParoQuant-quantized models in Section 5 with a single NVIDIA H200 GPU. We sample a training set of size 2048 evenly from WikiText2, C4, and RedPajama, and use 64 samples from Pile as the validation set to select the best parameters at each epoch. The training set and the validation set are shuffled with a fixed seed of 0. The sequence length of each sample is 2048. We set the batch size as 16, and apply a learning rate of 0.05 for rotation angles and channel-wise scaling, $10^{-5}$ for weights, and $10^{-6}$ for scales and zero points. The batch size and the learning rates are halved for the 70B model due to memory constraints. The rotation angles are initialized to 0, the channel-wise scaling factors are initialized to 1, and the scales and zero points are initialized using Equation (1). We use AdamW to optimize the parameters for 10 epochs at each stage (see Algorithm A2 for the two stages) with a cosine learning rate scheduler, which gradually decays the learning rate to $1/20$ of the original value. The hyperparameters weight_decay, betas, and eps of the AdamW optimizer are set to 0.01, (0.9, 0.95), and $10^{-10}$, respectively. We use SmoothL1Loss from PyTorch as the loss function.

## A.5 DECODING THROUGHPUT

Table A4 shows the decoding throughput of the original FP16 models and 4-bit models quantized with AWQ, QTIP, and ParoQuant. To ensure a fair comparison, we only replace the original linear

layers in the original models with the implementation provided by the official open-sourced repository of each baseline. For ParoQuant, we adopt the W4A16 GEMM kernels from the AWQ repository together with our transform kernel. All throughput results were obtained with PyTorch 2.6.0 using `torch.compile` in max-autotune mode and with CUDA Graphs enabled.

Table A4: Decoding (batch size = 1) throughput (tokens/s).

| RTX A6000 | Bits | Q3-1.7 | Q3-4 | L2-7 | L3-8 | Q3-8 | L2-13 | Q3-14 | Q3-32 | L3-70 |
|---|---|---|---|---|---|---|---|---|---|---|
| FP16 | 16 | 170 | 78 | 50 | 45 | 44 | 26 | 25 | OOM | OOM |
| AWQ | 4 | 320 | 176 | 140 | 120 | 113 | 78 | 70 | 34 | 17 |
| QTIP | 4 | 209 | 117 | 106 | 95 | 91 | 62 | 55 | 28 | 15 |
| PAROQ | 4 | 278 | 160 | 130 | 112 | 106 | 74 | 65 | 33 | 16 |
| **RTX 6000 Ada** | Bits | Q3-1.7 | Q3-4 | L2-7 | L3-8 | Q3-8 | L2-13 | Q3-14 | Q3-32 | L3-70 |
| FP16 | 16 | 213 | 99 | 63 | 56 | 55 | 33 | 31 | OOM | OOM |
| AWQ | 4 | 394 | 230 | 176 | 153 | 147 | 100 | 89 | 44 | 21 |
| QTIP | 4 | 270 | 166 | 138 | 125 | 118 | 83 | 80 | 40 | 20 |
| PAROQ | 4 | 341 | 206 | 163 | 142 | 136 | 94 | 84 | 42 | 21 |
| **RTX 4090** | Bits | Q3-1.7 | Q3-4 | L2-7 | L3-8 | Q3-8 | L2-13 | Q3-14 | Q3-32 | L3-70 |
| FP16 | 16 | 233 | 109 | 69 | 62 | 61 | OOM | OOM | OOM | OOM |
| AWQ | 4 | 433 | 251 | 192 | 167 | 159 | 109 | 98 | 48 | OOM |
| QTIP | 4 | 286 | 172 | 149 | 138 | 138 | 91 | 82 | 42 | OOM |
| PAROQ | 4 | 372 | 224 | 177 | 155 | 147 | 102 | 93 | 46 | OOM |

## A.6 TRAINING EFFICIENCY

Table A5 shows the calibration size and GPU time for quantizing LLaMA-3-8B on an NVIDIA H200 GPU. Although ParoQuant is slower than EfficientQAT due to an extra tuning stage and the additional computation graph nodes from independent rotations, it is significantly faster than QTIP, which requires significantly more calibration data and is slowed down by two extra steps in addition to layer-wise fine-tuning: generating Hessian matrices and end-to-end fine-tuning.

Table A5: Calibration data (# samples $\times$ sequence length) and GPU time for quantizing LLaMA-3-8B on an NVIDIA H200 GPU.

|  | AWQ | E-QAT | QTIP | PAROQ |
|---|---|---|---|---|
| Calibration Data | $128 \times 512$ | $4096 \times 2048$ | $4096 \times 8192$ | $2048 \times 2048$ |
| GPU Time | minutes | $\approx$ 3 hours | $\approx$ 20 hours | $\approx$ 9 hours |

## A.7 EVALUATION SETTINGS

**Perplexity Evaluation.** We use the test split in GPTQ to measure the perplexity on WikiText2 and C4. The sequence length is 8192 for LLaMA-3 and Qwen3 models, and 4096 for LLaMA-2 models. Note that the perplexity results of Qwen3 family in Table 1 are from the pre-trained "Base" models (*e.g.*, Qwen/Qwen3-8B-Base from Hugging Face), not the models after post-training (*e.g.*, Qwen/Qwen3-8B).

**Reasoning Task Evaluation.** We follow Liu et al. (2025a) and use Lighteval 0.8.1 (Habib et al., 2023) with vLLM 0.10.1 (Kwon et al., 2023) to evaluate the reasoning tasks in Table 2. We evaluate GPQA Diamond, AIME-24, and AIME-25 with three different seeds (42, 0, 1) and report the average accuracy to reduce variations, and evaluate MMLU-Pro with one seed (42).

**Non-Reasoning Task Evaluation.** We use the Language Model Evaluation Harness library (lm_eval) version 0.4.9.1 (Gao et al., 2024) to evaluate the tasks in Table 3 with the default settings of the library and a batch size of 32.

## A.8    USE OF LARGE LANGUAGE MODELS

The use of large language models for this work is limited to polishing the writing of the paper (e.g., checking grammatical errors, improving fluency, and enhancing clarity) and assisting with tasks that are not part of the core implementation (e.g., writing scripts for data visualization, plotting, or formatting results).

