# OpenReview forum: "ParoQuant: Pairwise Rotation Quantization for Efficient Reasoning LLM Inference"
_ICLR.cc/2026/Conference — ICLR 2026 Poster_

### Official Review · Reviewer_kgGQ · 2025-10-27

**Soundness:** 3
**Presentation:** 3
**Contribution:** 3
**Rating:** 8
**Confidence:** 4

**Summary:**

This paper presents ParoQuant, a weight-only post-training quantization (PTQ) method designed for reasoning LLMs. The core idea is to apply scaled pairwise rotation, combining independent Givens rotations with channel-wise scaling, to suppress outliers efficiently. The authors co-design a CUDA kernel to maintain high throughput. Experiments show consistent accuracy gains over AWQ and EfficientQAT, with less than 10% latency overhead.

**Strengths:**

The motivation that, quantization error accumulation in reasoning models, is clear and important.

The proposed rotation-based PTQ design is both novel and hardware-efficient.

Extensive experiments on multiple model sizes (up to 70B) and reasoning benchmarks (MMLU-Pro, GSM8K, etc.) demonstrate solid improvement.

Paper is clearly written and well-structured, with good algorithmic detail and ablations.

**Weaknesses:**

While the rotation kernel is claimed to be efficient, the paper lacks quantitative breakdown of runtime and memory overhead (e.g., FLOPs, memory traffic).

The scalability of independent rotations to larger group sizes or mixed-precision settings (e.g., W4A8) is not discussed.

More analysis on the trade-off between number of rotations and latency would strengthen the efficiency claim.

The method seems tailored for linear quantization; extension to vector quantization or activation quantization could be briefly discussed.

**Questions:**

Could the authors provide more detailed profiling on GPU resource usage and explain how ParoQuant scales when group size or model size further increases?

---

> ### Author Response · Authors · 2025-11-21
> **Response to Reviewer kgGQ (1/2)**
>
> Thank you very much for taking the time to review our work and for your thoughtful questions. We provide our detailed responses below.
>
> ---
>
> ***Q1: Quantitative breakdown of runtime and memory overhead (e.g., FLOPs, memory traffic).***
>
> The table below reports the latency (in μs) of different numbers of independent rotations for various channel dimensions on an RTX A6000 (batch size 1). As shown, 8 rotations incur roughly 2× the latency of memcpy, but remain lightweight relative to GEMM operations.
>
> | # rotations | dim=4096 | 8192 | 16384 | 32768 |
> | :--- | :--- | :--- | :--- | :--- |
> | 0 (memcpy) | 1.4 | 1.2 | 1.2 | 1.2 |
> | 1 | 1.6 | 1.6 | 1.5 | 1.6 |
> | 2 | 1.7 | 1.7 | 1.6 | 1.8 |
> | 4 | 1.9 | 1.9 | 1.9 | 2.1 |
> | 8 | 2.3 | 2.3 | 2.3 | 2.8 |
> | 16 | 3.2 | 3.2 | 3.2 | 4.2 |
>
> Memory traffic (load, in kB) scales linearly with the number of rotations. Values below include activations, rotation angles, pair indices, and channel-wise scales:
>
> | \# rotations | dim=4096 | 8192 | 16384 | 32768 |
> | :---- | :--- | :--- | :--- | :--- |
> | 0 (memcpy) | 16 | 33 | 66 | 131 |
> | 1 | 29 | 57 | 115 | 229 |
> | 2 | 41 | 82 | 164 | 328 |
> | 4 | 66 | 131 | 262 | 524 |
> | 8 | 115 | 229 | 459 | 918 |
> | 16 | 213 | 426 | 852 | 1704 |
>
> Given the RTX A6000’s memory bandwidth of 768 GB/s, the corresponding bandwidth utilization is:
>
> | \# rotations | dim=4096 | 8192 | 16384 | 32768 |
> | :--- | :--- | :--- | :--- | :--- |
> | 0 (memcpy) | 2.29% | 5.33% | 10.67% | 21.33% |
> | 1 | 3.00% | 6.00% | 12.80% | 24.00% |
> | 2 | 3.76% | 7.53% | 16.00% | 28.44% |
> | 4 | 5.05% | 10.11% | 20.21% | 36.57% |
> | 8 | 7.05% | 13.91% | 27.35% | 46.04% |
> | 16 | 9.00% | 18.00% | 36.00% | 54.86% |
>
> These results show that the rotation kernels are still well below full bandwidth saturation, even with 16 rotations.
>
> ---
>
> ***Q2: The scalability of independent rotations to larger group sizes or mixed-precision settings (e.g., W4A8).***
>
> The table below reports W4A16 results using a group size of 256. ParoQuant maintains the accuracy achieved at group size of 128 and the gap to AWQ widens at larger group sizes, indicating that reducing outliers becomes increasingly important in this setting.
>
> | Model | WikiText2 |  | C4 |  |
> | :---- | :---- | :---- | :---- | :---- |
> |  | ParoQuant | AWQ | ParoQuant | AWQ |
> | Llama-3-8B | **5.75** | 5.99 | **7.28** | 7.51 |
> | Qwen3-0.6B | **11.15** | 12.42 | **10.81** | 11.73 |
> | Qwen3-4B | **7.10** | 7.40 | **7.70** | 7.94 |
> | Qwen3-8B | **6.30** | 6.49 | **7.04** | 7.18 |
>
> Below are W4A8 results comparing ParoQuant (using RTN) and SpinQuant (based on GPTQ). Despite SpinQuant using a more accurate weight quantizer, ParoQuant achieves comparable or better performance.
>
> | Model | WikiText2 |  | C4 |  |
> | :---- | :---- | :---- | :---- | :---- |
> |  | ParoQuant (RTN) | SpinQuant (GPTQ)  | ParoQuant (RTN) | SpinQuant (GPTQ) |
> | Llama-3-8B | 5.92 | **5.84** | **7.41** | 7.42 |
> | Qwen3-0.6B | **11.40** | 11.84 | **10.96** | 11.47 |
> | Qwen3-4B | 7.19 | **7.18** | **7.76** | 7.81 |
> | Qwen3-8B | 6.37 | **6.35** | **7.08** | 7.10 |

---

> ### Author Response · Authors · 2025-11-21
> **Response to Reviewer kgGQ (2/2)**
>
> ***Q3: More analysis on the trade-off between number of rotations and latency.***
>
> We choose 8 independent rotations as a trade-off between expressiveness and runtime efficiency.
>
> The following table shows latency (µs) across different rotation counts on an RTX A6000:
>
> | # rotations | dim=4096 | 8192 | 16384 | 32768 |
> | :--- | :--- | :--- | :--- | :--- |
> | 0 (memcpy) | 1.4 | 1.2 | 1.2 | 1.2 |
> | 1 | 1.6 | 1.6 | 1.5 | 1.6 |
> | 2 | 1.7 | 1.7 | 1.6 | 1.8 |
> | 4 | 1.9 | 1.9 | 1.9 | 2.1 |
> | 8 | 2.3 | 2.3 | 2.3 | 2.8 |
> | 16 | 3.2 | 3.2 | 3.2 | 4.2 |
>
> Up to 8 rotations, the latency increase remains modest; 16 rotations incur substantially higher overhead.
>
> To evaluate expressiveness, we optimize various layers using different numbers of rotations and report the normalized reconstruction loss:
>
> | Layer (input dim, output dim) | RTN | Scaling | Scaling \+ 4R | Scaling \+ 8R | Scaling \+ 16R |
> | :---- | :---- | :---- | :---- | :---- | :---- |
> | Qwen3-0.6B q\_proj (1024, 1024\) | 5.37e-4 (100%)  | 4.21e-4 (78%) | 3.11e-4 (58%) | 2.84e-4 (53%) | 2.63e-4 (49%) |
> | Qwen3-0.6B gate\_proj (1024,3072) | 3.05e-3 (100%) | 2.3e-3 (75%) | 1.83e-3 (60%) | 1.68e-3 (55%) | 1.53e-3 (50%) |
> | Llama-3-8B k\_proj (4096, 1024\) | 1.34e-3 (100%) | 1.11e-3 (83%) | 3.87e-4 (29%) | 3.04e-4 (23%) | 2.43e-4 (18%) |
> | Llama-3-8B q\_proj (4096, 4096\) | 6.33e-4 (100%) | 5.14e-4 (81%) | 1.79e-4 (29%)  | 1.38e-4 (22%) | 1.19e-4 (19%) |
> | Llama-3-8B gate\_proj (4096, 14336\) | 7.12e-5 (100%) | 6.97e-5 (98%) | 6.47e-5 (91%) | 6.31e-5 (89%) | 6.13e-5 (86%) |
> | Qwen3-14B q\_proj (5120, 5120\) | 3.71e-6 (100%) | 3.57e-6 (96%) | 3.49e-6 (94%) | 3.43e-6 (92%) | 3.38e-6 (91%) |
> | Qwen3-14B gate\_proj (5120, 17408\) | 1.23e-4 (100%) | 8.39e-5 (68%)  | 6.51e-5 (53%) | 6.25e-5 (51%) | 5.90e-5 (48%) |
>
> The improvements from 8 → 16 rotations are small, while 4 → 8 rotations still yield meaningful gains in important layers such as q_proj. Therefore, 8 rotations provide the best balance between expressiveness and efficiency.
>
> ---
>
> ***Q4: The method seems tailored for linear quantization; extension to vector quantization or activation quantization could be briefly discussed.***
>
> **Vector quantization:** The scaled pairwise rotation transform is compatible with any differentiable quantizer, including trellis-based quantizers such as QTIP. Since the transform effectively suppresses outliers, similar in spirit to the incoherence processing in QTIP and QuIP#, it is expected to benefit vector quantizers and outperform Hadamard-based transforms, as discussed in Appendix A.2.
>
> **Activation quantization:** We evaluated ParoQuant under W4A4 (INT) settings using the same experimental setup described in the paper, without tuning any hyperparameters.
>
> Under W4A4, ParoQuant consistently outperforms SpinQuant, often by a substantial margin in perplexity:
>
> | Model | WikiText2 |  | C4 |  |
> | :---- | :---- | :---- | :---- | :---- |
> |  | ParoQuant (RTN) | SpinQuant (GPTQ) | ParoQuant (RTN) | SpinQuant (GPTQ) |
> | Llama-3-8B | **6.21** | 6.77 | **7.69** | 8.31 |
> | Qwen3-0.6B | **12.45** | 15.94 | **11.73** | 14.68 |
> | Qwen3-4B | **7.51** | 7.97 | **8.03** | 8.55 |
> | Qwen3-8B | 7.18 | **7.13** | **7.68** | 7.89 |
>
> The advantage is even more pronounced on reasoning benchmarks:
>
> | DeepSeek-Distill-Llama-8B | AIME-24 | AIME-25 | GSM8K | MATH-500 | GPQA-Diamond |
> | :---- | :---- | :---- | :---- | :---- | :---- |
> | FP16 | 48.89 | 26.67 | 88.78 | 89.93 | 48.65 |
> | SpinQuant | 14.44 | 17.78 | 80.11 | 75.40 | 33.50 |
> | ParoQuant | **30.00** | **22.22** | **86.53** | **85.27** | **42.93** |
>
> These results highlight the robustness of ParoQuant under activation quantization.
>
> ***Q5: How does ParoQuant scale when group size or model size further increases?***
>
> We evaluate the latency of 8 independent rotations across group sizes 128, 256, and 512 (RTX A6000, batch size of 1). Larger group sizes introduce only small additional overhead, with negligible effect on end-to-end throughput.
>
> | group size | dim=4096 | 8192 | 16384 | 32768 |
> | :--- | :--- | :--- | :--- | :--- |
> | 128 | 2.27 | 2.30 | 2.34 | 2.78 |
> | 256 | 2.59 | 2.56 | 2.53 | 2.91 |
> | 512 | 3.30 | 3.10 | 3.00 | 3.14 |
>
> Throughput results for larger models are provided in Appendix A.4 (L814-839). The relative overhead of ParoQuant decreases as model size grows because the rotation operations are lightweight compared to GEMM, and thus amortized more effectively in larger layers.
>
> ---
>
> We hope these responses fully address your questions. Thank you again for your constructive feedback! We welcome any further discussion or suggestions.

---

> > ### Comment · Reviewer_kgGQ · 2025-11-28
> > **Response to the authors**
> >
> > Thank you for your response. I choose to keep my rating.

---

### Official Review · Reviewer_xZZn · 2025-10-30

**Soundness:** 3
**Presentation:** 3
**Contribution:** 3
**Rating:** 6
**Confidence:** 5

**Summary:**

The paper presents ParoQuant, a weight-only post-training quantization method developed to improve the accuracy and efficiency of LLMs. It tackles the challenge of quantization error through a novel combination of independent Givens rotations and channel-wise scaling, which effectively reduces the impact of outliers. Additionally, the work incorporates a custom CUDA kernel to accelerate online scaled pairwise rotations, enabling faster inference.

**Strengths:**

1. Clear and Well-Founded Motivation: The paper observes that rotating only the top 10% of the most significant weight channel pairs can achieve nearly the same reduction in quantization error as performing a full rotation. This insight eliminates a large amount of redundant computation from full matrix multiplications, leading to a much more efficient quantization process.

2. Methodology with GPU-Aware Design: Building on this motivation, the authors propose a three-step design for the scaled pairwise rotation transform.

Step 1: Replace costly full orthogonal matrix multiplications with a set of decomposed Givens rotations.

Step 2: Eliminate inter-rotation dependencies to allow fully parallel execution on GPUs, resulting in independent rotations.

Step 3: Since a single independent rotation cannot adequately capture complex weight distributions, apply a series of independent rotations combined with channel-wise scaling to improve representation and quantization accuracy.

3. Practical CUDA Implementation: The paper further introduces a co-designed efficient transform kernel that maximizes GPU parallelism by executing computations across three levels:

Token-level: Parallelization over the token dimension of the activation tensor.

Channel-group level: Different CUDA blocks handle different groups of channels.

Pair level: Each rotation pair is processed by a separate CUDA thread.

**Weaknesses:**

Overall, I found the paper well-written and technically solid. The following are just minor curiosities rather than critical weaknesses:

1. The 4-bit performance gains appear somewhat modest for certain model sizes and tasks (e.g., Perplexity and AIME). It would be interesting to see whether ParaQuant delivers more substantial improvements at lower bitwidths, such as 3-bit or 2-bit quantization.

2. Do you have any insight into why E-QAT performs particularly poorly on AIME, given that ParaQuant can essentially be viewed as an extension of E-QAT with additional learnable rotations? I am expecting the performance gap between the two methods to be smaller.

**Questions:**

See Above

---

> ### Author Response · Authors · 2025-11-21
> **Response to Reviewer xZZn**
>
> Thank you very much for taking the time to review our work and for your thoughtful questions. We provide our detailed responses below.
>
> ---
>
> ***Q1: It would be interesting to see whether ParaQuant delivers more substantial improvements at lower bitwidths, such as 3-bit or 2-bit quantization.***
>
> Below we report results for ParoQuant and AWQ under 3-bit (group size 128) and 2-bit (group size 64) weight quantization. As expected, the performance gap widens as the bitwidth decreases. ParoQuant provides consistently larger gains over AWQ in 3-bit settings, and at 2-bit, where AWQ collapses, ParoQuant remains stable and significantly more accurate.
>
> Here are W3A16 results with group size of 128:
>
> | Model | WikiText2 |  | C4 |  |
> | :---- | :---- | :---- | :---- | :---- |
> |  | ParoQuant | AWQ | ParoQuant | AWQ |
> | Llama-3-8B | **6.54** | 7.38 | **7.94** | 8.8 |
> | Qwen3-0.6B | **12.88** | 18.63 | **12.1** | 16.28 |
> | Qwen3-4B | **7.68** | 8.68 | **8.17** | 8.96 |
> | Qwen3-8B | **6.75** | 7.28 | **7.4** | 7.77 |
>
> And here are W2A16 results with group size of 64:
>
> | Model | WikiText2 |  | C4 |  |
> | :---- | :---- | :---- | :---- | :---- |
> |  | ParoQuant | AWQ | ParoQuant | AWQ |
> | Llama-3-8B | **21.25** | 1.78e6 | **18.86** | 2.10e6 |
> | Qwen3-0.6B | **55.7** | 1.34e8 | **35.01** | 8.95e7 |
> | Qwen3-4B | **16.7** | 3.87e7 | **13.76** | 2.93e7 |
> | Qwen3-8B | **11.49** | 2.26e9 | **10.65** | 2.75e9 |
>
> These results show that ParoQuant is especially robust in ultra-low-bit settings, where AWQ struggles or fails.
>
> ---
>
> ***Q2: Do you have any insight into why E-QAT performs particularly poorly on AIME?***
>
> Although ParoQuant builds on E-QAT’s idea of jointly fine-tuning weights and quantization parameters, the nature of the weight updates differs substantially. ParoQuant applies optimization **after transforming the weights to remove major outliers**, meaning the subsequent updates operate on **outlier-reduced weights** and only need to correct a few remaining isolated outliers.
>
> In contrast, E-QAT performs optimization directly on the **original weight space**, which contains **many large outliers**. As a result, its weight updates must be much larger to compensate for these outliers, making the model far more susceptible to **overfitting**.
>
> This behavior aligns with our empirical observations: E-QAT improves perplexity significantly over AWQ, but provides only marginal gains on non-reasoning benchmarks and underperforms on long-horizon reasoning tasks, particularly on AIME, the hardest evaluation (*we have updated the table with more stable results*). We believe that this overfitting induced by optimizing outlier-heavy weights is the main reason for E-QAT’s poor AIME performance. ParoQuant mitigates this issue by first smoothing the weight distribution, resulting in more stable and generalizable updates.
>
> ---
>
> We hope these responses fully address your questions. Thank you again for your constructive feedback! We welcome any further discussion or suggestions.

---

### Official Review · Reviewer_hpqW · 2025-10-30

**Soundness:** 3
**Presentation:** 3
**Contribution:** 3
**Rating:** 8
**Confidence:** 5

**Summary:**

This paper introduces ParoQuant, a novel weight-only PTQ method for reasoning LLMs. It uses a "scaled pairwise rotation" transform, combining channel-wise scaling with hardware-efficient Givens rotations to suppress outliers. Through algorithm-system co-design, it achieves high accuracy with low inference overhead. It provides a very systematic solution, and the experiments are very comprehensive.

**Strengths:**

1.	The paper clearly identifies and addresses a critical, forward-looking problem: the poor performance of efficient quantization methods on reasoning tasks that require long chains of thought. This focus on error accumulation in generative tasks is timely and important.

2.	The proposed "scaled pairwise rotation" is a novel and elegant solution. The insight that a full rotation matrix is redundant and can be effectively approximated by a series of independent, parallelizable Givens rotations is the key contribution and is very well executed.

3.	The algorithm-system co-design is a major strength. The authors didn't just propose a transform; they designed a custom CUDA kernel that makes the transform viable in practice, demonstrating a deep understanding of both the algorithmic and hardware constraints. The empirical results, showing ParoQuant matching QTIP's accuracy while being ~25% faster, are very compelling.

**Weaknesses:**

1.	The greedy pair selection strategy outlined in Algorithm A1, while effective and intuitive, may not be globally optimal. It would be beneficial for the authors to discuss the potential limitations of this greedy approach.

2.	In Section 3, when discussing quantization degradation on reasoning tasks, the authors should cite other recent works that have also identified this specific problem (e.g., QSPEC) to better contextualize their motivation.

3.	In Figure 3, some text labels in the right-most portion of the diagram are overlapped, which slightly hinders readability.

**Questions:**

1.	The number of independent rotations is fixed at K=8 for most experiments. How was this number chosen? Is there a clear point of diminishing returns, and does the optimal value of K change depending on the model architecture or size?

2.	The "pairwise" rotation in Algorithm A1 is effective but seems conservative. Did the authors consider more fine-grained or alternative rotation structures, such as rotating small blocks of channels against each other? While this might be more expressive, it would likely introduce significant scheduling overhead. A discussion on this potential trade-off between rotation granularity and scheduling efficiency would be insightful.

---

> ### Author Response · Authors · 2025-11-21
> **Response to Reviewer hpqW (1/2)**
>
> Thank you very much for taking the time to review our work and for your thoughtful questions. We provide our detailed responses below.
>
> ---
>
> ***Q1: The greedy pair selection strategy outlined in Algorithm A1, while effective and intuitive, may not be globally optimal.***
>
> We agree that the initial pair selection algorithm in A1 is not optimal. However, both theoretical analysis and empirical evidence show that a sequence of independent rotations is highly robust to the pair selection strategy. In practice, greedy selection or even random pairing is **nearly optimal** for our primary goal: pairing outlier channels with non-outlier channels to suppress activation outliers.
>
> To support this, we provide a theoretical analysis of random pair selection. Consider a group of $N$ channels with $n$ outliers. In one independent rotation, the probability that every outlier channel is paired with a non-outlier channel is:
>
> $$
> P(n)=\frac{N-n}{N-1} \frac{N-n-1}{N-3} \cdots \frac{N-2n+1}{N-2n+1}
> = \prod_{k=0}^{n-1}\frac{N-n-k}{N-1-2k},
> $$
>
> After $K$ independent rotations, the probability that all outlier channels have been paired with non-outlier channels at least once is:
>
> $$P\ge 1-(1-P(n))^K.$$
>
> With our configuration of $N=128$ and $K=8$, the resulting probabilities are:
>
> | $n$ | % of channels | Probability |
> | :---- | :---- | :---- |
> | 5 | 4% | 1.0 |
> | 10 | 8% | 1.0 |
> | 15 | 12% | 0.98 |
> | 20 | 16% | 0.78 |
>
> Therefore, for realistic outlier counts (<10%), random pairing already guarantees near-optimal pairing with very high probability. Although the original design used a greedy strategy, our updated implementation adopts random selection for simplicity, and our experiments on perplexity and downstream reasoning tasks confirm that this change does not affect end-to-end performance.
>
> ---
>
> ***Q2: The authors should cite other recent works that have also identified this specific problem to better contextualize their motivation.***
>
> Thank you for the suggestion. We have updated the introduction to include two recent works, QSPEC [1] and [2], that support our observations on the degradation of reasoning ability under quantization:
>
> [1] Zhao et al., *QSPEC: Speculative decoding with complementary quantization schemes*.
>
> [2] Liu et al., *Quantization hurts reasoning? an empirical study on quantized reasoning models*.
>
> ---
>
> ***Q3: In Figure 3, some text labels in the right-most portion of the diagram are overlapped.***
>
> We have updated the figure to resolve the readability concerns.

---

> ### Author Response · Authors · 2025-11-21
> **Response to Reviewer hpqW (2/2)**
>
> ***Q4: The number of independent rotations is fixed at K=8 for most experiments. How was this number chosen? Is there a clear point of diminishing returns, and does the optimal value of K change depending on the model architecture or size?***
>
> We choose 8 independent rotations as a trade-off between expressiveness and runtime efficiency.
>
> The following table shows latency (µs) across different rotation counts on an RTX A6000:
>
> | # rotations | dim=4096 | 8192 | 16384 | 32768 |
> | :--- | :--- | :--- | :--- | :--- |
> | 0 (memcpy) | 1.4 | 1.2 | 1.2 | 1.2 |
> | 1 | 1.6 | 1.6 | 1.5 | 1.6 |
> | 2 | 1.7 | 1.7 | 1.6 | 1.8 |
> | 4 | 1.9 | 1.9 | 1.9 | 2.1 |
> | 8 | 2.3 | 2.3 | 2.3 | 2.8 |
> | 16 | 3.2 | 3.2 | 3.2 | 4.2 |
>
> Up to 8 rotations, the latency increase remains modest; 16 rotations incur substantially higher overhead.
>
> To evaluate expressiveness, we optimize various layers using different numbers of rotations and report the normalized reconstruction loss:
>
> | Layer (input dim, output dim) | RTN | Scaling | Scaling \+ 4R | Scaling \+ 8R | Scaling \+ 16R |
> | :---- | :---- | :---- | :---- | :---- | :---- |
> | Qwen3-0.6B q\_proj (1024, 1024\) | 5.37e-4 (100%)  | 4.21e-4 (78%) | 3.11e-4 (58%) | 2.84e-4 (53%) | 2.63e-4 (49%) |
> | Qwen3-0.6B gate\_proj (1024,3072) | 3.05e-3 (100%) | 2.3e-3 (75%) | 1.83e-3 (60%) | 1.68e-3 (55%) | 1.53e-3 (50%) |
> | Llama-3-8B k\_proj (4096, 1024\) | 1.34e-3 (100%) | 1.11e-3 (83%) | 3.87e-4 (29%) | 3.04e-4 (23%) | 2.43e-4 (18%) |
> | Llama-3-8B q\_proj (4096, 4096\) | 6.33e-4 (100%) | 5.14e-4 (81%) | 1.79e-4 (29%)  | 1.38e-4 (22%) | 1.19e-4 (19%) |
> | Llama-3-8B gate\_proj (4096, 14336\) | 7.12e-5 (100%) | 6.97e-5 (98%) | 6.47e-5 (91%) | 6.31e-5 (89%) | 6.13e-5 (86%) |
> | Qwen3-14B q\_proj (5120, 5120\) | 3.71e-6 (100%) | 3.57e-6 (96%) | 3.49e-6 (94%) | 3.43e-6 (92%) | 3.38e-6 (91%) |
> | Qwen3-14B gate\_proj (5120, 17408\) | 1.23e-4 (100%) | 8.39e-5 (68%)  | 6.51e-5 (53%) | 6.25e-5 (51%) | 5.90e-5 (48%) |
>
> The improvements from 8 → 16 rotations are small, while 4 → 8 rotations still yield meaningful gains in important layers such as q_proj. Therefore, 8 rotations provide the best balance between expressiveness and efficiency.
>
> ---
>
> ***Q5: Did the authors consider more fine-grained or alternative rotation structures, such as rotating small blocks of channels against each other?***
>
> We appreciate the suggestion of using block-wise rotations. Let $N$ be the dimension and $n$ the block size. A block-wise rotation contains $\frac{N}{n} \cdot \frac{1}{2}n(n-1)=\frac{N}{2}\cdot (n-1)$ parameters, while $K$ independent rotations contain $\frac{N}{2}\cdot K$ parameters. A fair comparison requires $n-1=K$, so we compare block size 8 to 7 independent rotations.
>
> We evaluate layers 0, 15, and 31 of Llama-3-8B and report the average final loss normalized to RTN:
>
> | Layer | RTN | 7 IR (gs 128\) | Block-wise (gs 8\) |
> | :---- | :---- | :---- | :---- |
> | self\_attn.k\_proj | 100% | 44.04% | 51.60% |
> | self\_attn.v\_proj | 100% | 66.72% | 68.94% |
> | self\_attn.q\_proj | 100% | 46.25% | 50.98% |
> | self\_attn.o\_proj | 100% | 91.34% | 91.54% |
> | mlp.up\_proj | 100% | 92.91% | 92.56% |
> | mlp.gate\_proj | 100% | 90.46% | 90.48% |
>
> For layers with few outliers (o_proj, up_proj, gate_proj), both approaches perform similarly. For layers with many outliers (k_proj, v_proj, q_proj), block-wise rotations are less effective because the block size (8) does not align with the quantization group size (128), limiting their ability to suppress outliers.
>
> Independent rotations maintain fixed parameters and compute cost with respect to the number of rotations, independent of the group size, making them more suitable for large quantization groups such as 128.
>
> ---
>
> We hope these responses fully address your questions. Thank you again for your constructive feedback! We welcome any further discussion or suggestions.

---

### Official Review · Reviewer_7v1a · 2025-10-31

**Soundness:** 3
**Presentation:** 3
**Contribution:** 3
**Rating:** 6
**Confidence:** 3

**Summary:**

This paper introduces ParoQuant, a weight-only post-training quantization (PTQ) method designed for reasoning LLMs, where quantization errors can accumulate over long generations.
ParoQuant combines: 1.Independent Givens rotations (pairwise rotations) to suppress outliers efficiently, and 2.Channel-wise scaling to even out magnitude across channels.

**Strengths:**

The authors convincingly argue that reasoning LLMs are especially sensitive to accumulated quantization errors, providing strong justification for the proposed method’s focus on accuracy stability during long generation.

ParoQuant achieves higher reasoning-task accuracy than AWQ and matches the state-of-the-art QTIP while being significantly faster.

The paper thoughtfully co-designs the quantization algorithm and CUDA implementation.

**Weaknesses:**

Please see my questions.

**Questions:**

How does ParoQuant perform under activation quantization or mixed-precision scenarios?

Could the pairwise rotation be merged offline to further reduce runtime cost?

Can this idea extend to FP4/FP8 formats?

---

> ### Author Response · Authors · 2025-11-21
> **Response to Reviewer 7v1a**
>
> Thank you very much for taking the time to review our work and for your thoughtful questions. We provide our detailed responses below.
>
> ---
>
> ***Q1: How does ParoQuant perform under activation quantization or mixed-precision scenarios?***
>
> We evaluated ParoQuant under W4A4 (INT) and W4A8 (INT) settings using the same experimental setup described in the paper, without tuning any hyperparameters.
>
> **Under W4A4**, ParoQuant consistently outperforms SpinQuant, often by a substantial margin in perplexity:
>
> | Model | WikiText2 |  | C4 |  |
> | :---- | :---- | :---- | :---- | :---- |
> |  | ParoQuant (RTN) | SpinQuant (GPTQ) | ParoQuant (RTN) | SpinQuant (GPTQ) |
> | Llama-3-8B | **6.21** | 6.77 | **7.69** | 8.31 |
> | Qwen3-0.6B | **12.45** | 15.94 | **11.73** | 14.68 |
> | Qwen3-4B | **7.51** | 7.97 | **8.03** | 8.55 |
> | Qwen3-8B | 7.18 | **7.13** | **7.68** | 7.89 |
>
> The advantage is even more pronounced on reasoning benchmarks:
>
> | DeepSeek-Distill-Llama-8B | AIME-24 | AIME-25 | GSM8K | MATH-500 | GPQA-Diamond |
> | :---- | :---- | :---- | :---- | :---- | :---- |
> | FP16 | 48.89 | 26.67 | 88.78 | 89.93 | 48.65 |
> | SpinQuant | 14.44 | 17.78 | 80.11 | 75.40 | 33.50 |
> | ParoQuant | **30.00** | **22.22** | **86.53** | **85.27** | **42.93** |
>
> **Under W4A8**, ParoQuant matches or outperforms SpinQuant:
>
> | Model | WikiText2 |  | C4 |  |
> | :---- | :---- | :---- | :---- | :---- |
> |  | ParoQuant (RTN) | SpinQuant (GPTQ) | ParoQuant (RTN) | SpinQuant (GPTQ) |
> | Llama-3-8B | 5.92 | **5.84** | **7.41** | 7.42 |
> | Qwen3-0.6B | **11.40** | 11.84 | **10.96** | 11.47 |
> | Qwen3-4B | 7.19 | **7.18** | **7.76** | 7.81 |
> | Qwen3-8B | 6.37 | **6.35** | **7.08** | 7.10 |
>
> These results highlight the robustness of ParoQuant under activation quantization and mixed-precision settings.
>
> ---
>
> ***Q2: Can the pairwise rotation be merged offline to further reduce runtime cost?***
>
> Yes, part of the rotation can be merged offline. However, the benefit is limited: only the rotation applied to the output projection of the self-attention block is mergeable. Since a decoder layer contains seven linear layers, this reduces the rotation overhead by only about **1/7**, resulting in **<3% end-to-end throughput improvement** on an RTX A6000.
>
> |  | Original | Merged |
> | :--- | :--- | :--- |
> | Q3-1.7 | 278 | 284 |
> | Q3-4 | 160 | 165 |
> | L2-7 | 130 | 131 |
> | L3-8 | 112 | 113 |
> | Q3-8 | 106 | 107 |
> | L2-13 | 74 | 74 |
> | Q3-14 | 65 | 67 |
> | Q3-32 | 33 | 33 |
> | L3-70 | 16.3 | 16.5 |
>
> ---
>
> ***Q3: Can this idea extend to FP4/FP8 formats?***
>
> Yes! ParoQuant naturally extends to FP4/FP8 quantization.
>
> Below are results for **MXFP4**, where ParoQuant significantly outperforms RTN:
>
> | Model | WikiText2 |  | C4 |  |
> | :---- | :---- | :---- | :---- | :---- |
> |  | ParoQuant | RTN | ParoQuant | RTN |
> | Llama-3-8B | **6.30** | 7.63 | **7.76** | 8.90 |
> | Qwen3-0.6B | **13.6** | 17.6 | **12.48** | 15.94 |
> | Qwen3-4B | **7.57** | 10.36 | **8.10** | 10.38 |
> | Qwen3-8B | **6.91** | 7.64 | **7.56** | 8.27 |
>
> And similarly for **NVFP4**:
>
> | Model | WikiText2 |  | C4 |  |
> | :---- | :---- | :---- | :---- | :---- |
> |  | ParoQuant | RTN | ParoQuant | RTN |
> | Llama-3-8B | **6.05** | 6.18 | **7.54** | 7.67 |
> | Qwen3-0.6B | **12.47** | 13.75 | **11.80** | 12.74 |
> | Qwen3-4B | **7.44** | 7.65 | **7.98** | 8.17 |
> | Qwen3-8B | **6.54** | 6.71 | **7.24** | 7.40 |
>
> ---
>
> We hope these responses fully address your questions. Thank you again for your constructive feedback! We welcome any further discussion or suggestions.

---

### Author Response · Authors · 2025-11-21
**General Response**

We sincerely appreciate the reviewers’ thoughtful and encouraging feedback. Across all four reviews, several strengths of our work were consistently highlighted:
- **Clear motivation:** a timely focus on the unique sensitivity of reasoning LLMs to quantization error accumulation.
- **Methodological novelty:** the scaled pairwise rotation transform, with the insight that independent Givens rotations can efficiently approximate full rotations.
- **Strong algorithm-system co-design:** an efficient CUDA kernel that enables practical deployment with low runtime overhead.
- **Comprehensive evaluation:** robust accuracy improvements and low overhead across models of many scales, particularly on long-chain reasoning tasks.

We are grateful for these positive assessments. In response to the reviewers’ suggestions, we have made several revisions and clarifications. All main conclusions of the paper remain unchanged.

**1. Simplified pair selection (Algorithm A1) and updated results**
We replaced the original greedy pairing strategy with random selection. Our theoretical analysis and new experiments show that random pairing performs comparably to greedy pairing while simplifying the PTQ pipeline. All ParoQuant results have been updated accordingly (see response to reviewer *hpqW*).

**2. Added QTIP results on the Qwen3 family**
We extended QTIP comparisons to the Qwen3 models, following the calibration setup used in the QTIP paper for Llama-3 to ensure consistency.

**3. Unified evaluation of QTIP and EfficientQAT**
To ensure fair comparisons, we now evaluate both methods using their pseudo-quantized FP16 models with the same unified evaluation script as ParoQuant, rather than relying on their original real-quantized inference pipelines.

**4. Improved reasoning-task evaluation**
Following best practices from Liu et al. [1], we refined our evaluation pipeline for reasoning tasks:
- Adopted LightEval [2], which provides stronger reasoning-task support than lm-eval [3].
- Increased the generation limit from 8k to 32k tokens to avoid truncation.
- Added results for Qwen3-4B and MMLU-Pro for DeepSeek-R1-Distill-Llama-8B.
- For GPQA-Diamond, AIME-24, and AIME-25, we now report averages over 3 seeds.
- GSM8K was removed due to its short-answer format and limited diagnostic value for long-chain reasoning.

**5. Extension to weight–activation quantization (W4A4, W4A8)**
As requested by reviewers *7v1a* and *kgGQ*, we extended ParoQuant to **weight-activation quantization**, including INT4/INT8 and FP4 formats (MXFP4, NVFP4). Across all mixed-precision settings, ParoQuant consistently outperforms baseline methods. Additional results are provided in our response to reviewer *7v1a*.

---

[1] Liu et al. *Quantization hurts reasoning? An empirical study on quantized reasoning models.*
[2] https://github.com/huggingface/lighteval
[3] https://github.com/EleutherAI/lm-evaluation-harness

---

### Meta-Review · Area_Chair_Xxhw · 2026-01-02

**Summary:**

**Paper Summary:**
ParoQuant introduces a weight-only post-training quantization method for reasoning LLMs that combines scaled pairwise rotations and channel-wise scaling with a GPU-efficient implementation to reduce quantization errors and improve inference accuracy.

**Strengths:**

1. Clear motivation: Addresses the unique sensitivity of reasoning LLMs to quantization error accumulation over long chains of thought.
2. Novel methodology: Introduces scaled pairwise rotation using independent Givens rotations and channel-wise scaling to suppress outliers efficiently.
3. Algorithm–system co-design: Implements a custom CUDA kernel for practical deployment with low runtime overhead.
4. Comprehensive evaluation: Demonstrates consistent accuracy improvements over AWQ and competitive performance with QTIP across multiple model sizes and reasoning benchmarks.
5. Robustness: Extends effectively to mixed-precision and ultra-low-bit settings, maintaining stability where baselines fail.

**Weaknesses:**

1. Limited discussion on scalability to larger group sizes and mixed-precision settings in the original text (later addressed in author response).
2. Lack of detailed profiling of runtime and memory overhead in the main paper.
3. Greedy pair selection strategy initially presented may not be globally optimal.
4. Performance gains at 4-bit quantization appear modest for some tasks; lower bitwidth results were not initially included.
5. Method seems tailored for linear quantization; extensions to vector quantization and activation quantization were only briefly discussed.

**Reviewer Concerns:**

**Reviewer 7v1a:**
I think most concerns have been well addressed.

**Reviewer hpqW:**

The concerns about pair selection, citations, figure fix, and rotation trade-off have been addressed.
The concern about alternative rotation structures is partially addressed.

**Reviewer xZZn:**
The concern of lower bitwidth and E-QAT explanation has been addressed.
The concern of modest 4-bit gains is still outstanding.

**Reviewer kgGQ:**
The concerns of runtime/memory, scalability, trade-off, activation quantization have been addressed.
The concern of GPU resource profiling is partially addressed.

**Reviewer Scores:**

The paper already received high scores. Although the authors have addressed most of the review comments, I am not sure whether the reviewers will further raise their scores.

---

### Decision · Program_Chairs · 2026-01-26

Accept (Poster)